# Computational Modelling of the Impacts of Saltmarsh Management Interventions on Hydrodynamics of a Small Macro-Tidal Estuary

**William G. Bennett [1],\*** , **Thomas J. van Veelen [1]**, **Tom P. Fairchild [2]** , **John N. Griffin [2]** and **Harshinie Karunarathna [1]**

[1] College of Engineering, Swansea University, Swansea SA1 8EN, UK; thomas.vanveelen@swansea.ac.uk (T.J.v.V.); h.u.karunarathna@swansea.ac.uk (H.K.)

[2] Department of Biosciences, Swansea University, Swansea SA2 8PP, UK; tom.phillip.fairchild@googlemail.com (T.P.F.); j.n.griffin@swansea.ac.uk (J.N.G.)

\* Correspondence: w.g.bennett@swansea.ac.uk; Tel.: +44-1792-604222

**Abstract:** Saltmarshes are considered as natural coastal defences. However, owing to the large context dependency, there is much discussion over their effectiveness in providing coastal protection and the necessity of additional coastal defence interventions. The macro-tidal Taf Estuary in south-west Wales was chosen as the case study in this paper to investigate the effects of anthropogenic coastal defence interventions such as construction of hard defences, managed realignment, and altering land use of the saltmarshes on the complex hydrodynamics of the estuary. A coupled flow–wave–vegetation model, developed using the Delft3D coastal modelling software, was used. The wave and current attenuation role of saltmarshes during two contrasting storm conditions was modelled, with and without saltmarsh management interventions. The study reveals that certain saltmarsh management interventions can have widespread impacts on the hydrodynamics of the estuary. Altering the land use by allowing extensive grazing of saltmarsh by livestock was found to have the largest impact on wave attenuation, where wave heights on the marsh almost doubled when compared with the no-intervention scenario. On the other hand, managed realignment has a significant impact on tidal currents, where tidal currents reached 0.5 m/s at certain locations. Changes in estuarine hydrodynamics can lead to undesired impacts on flooding and erosion, which stresses the importance of understanding the effects of localized anthropogenic coastal management interventions on the entire estuarine system.

**Keywords:** coastal management; saltmarsh; estuarine hydrodynamics; Delft3D; computational modelling

## 1. Introduction

The sustainability of coastal defence practice in the United Kingdom and elsewhere is under increasing scrutiny owing to growing costs and the implications of global climate change. The combination of more natural and less engineered approaches, which are known as 'nature-based coastal defence approaches', has been identified as a desirable solution as opposed to hard coastal defences alone. Saltmarshes have been found to act as natural buffer zones, providing protection from storms and flooding [1]. In addition, saltmarshes can help reduce vulnerability of coastal defence structures to bed scour and erosion [2], and lessen the effects of wave action on the structure [3]. Compared with mud flats, saltmarshes have been demonstrated to be more effective at providing wave damping [3–5]. Thus, a saltmarsh combined with a coastal defence may reduce the size and height of a seawall required to manage a coastline, with a near-linear relationship between saltmarsh width and

seawall height for comparable levels of protection [6–8]. Saltmarsh vegetation has also been shown to reduce wave setup [9] and alter the flow velocity profiles within the marsh and beyond [10–12]. These effects in turn can alter sediment dynamics, thus contributing to coastal morphodynamic evolution [11–13].

Saltmarshes have historically suffered and continue to suffer from massive loss of area and alteration of saltmarsh habitats worldwide [14,15] as a result of numerous factors. These include global climate change driven sea level rise, exploitation of marshes as grazing grounds or for other resources, shortage of sediment supply owing to damming of rivers or reduced littoral transport, and hard coastal defences that were built mostly without due consideration of the interaction between saltmarsh morphology and ecology (e.g., Doody [16]). The complex interactions and feedbacks between the hydrodynamics, vegetation, and morphodynamics are key to saltmarsh resilience against these factors, depending on the robustness of the marsh to its new surroundings [17].

Shoreline management plans (SMPs) are used within the United Kingdom coastal defence management planning, providing large-scale assessment of risks, and the necessary measures to address them. On the basis of observations and modelling, the most recent SMPs indicate that losses in saltmarshes could be substantial in the medium- to long-term future with increasing rates of sea level rise [18–20]. Current U.K. coastal management policy aims to achieve no net loss in habitat area, which puts pressure on coastal management to consider a range of options, such as the rollback of existing defences and allowing for land to compensate for areas lost to the sea. This managed realignment approach is recognised as a relatively new "soft" engineering technique when compared with existing widely adopted techniques [21]. It involves creating new intertidal areas from formerly flood-defended areas of coastal land to provide sustainable flood defences, new intertidal habitats, or a combination of those. Compared with hard coastal defences, it is often a cheaper alternative when compared with the expenses involved in upgrading or repairing hard defences [21]. While public opinion may view managed realignment as representing "giving in" to the sea, it is becoming increasingly utilised in the United Kingdom, with approximately fifty different sites already completed [18,22,23] There are further plans in place to realign 10% of the English and Welsh coasts, thus creating 62 km$^2$ of coastal land by 2030 and 115 km$^2$ by 2060 [24,25].

At managed realignment sites around the United Kingdom, a range of impacts have been observed, owing to differing influencing factors. At Tollesbury on the south east of the U.K. coastline, the landward realignment of coastal defences quickly produced intertidal mudflats on low-lying agricultural land [26]. While at Freiston shore, as well as on the east coast of the United Kingdom, higher than predicted ebb tides led to damage to oyster fishery [2]. Moreover, the creation of the realignment site at Freiston impacted the wider saltmarsh as well as the realigned area itself [27,28]. Friess et al. [28] noted that managed realignment within estuaries provides a significant sudden perturbation to the hydrodynamic equilibrium that may exist between the flood and ebb tidal currents for the existing morphology. Using a computational model of the Humber estuary, Townend and Pethick [29] showed that large areas of managed realignment in the Humber could reduce the height of extreme high water events. Further to this, Pontee [30] found that the realignment scheme design can lead to differences in the impacts on hydrodynamic behaviour, with large realignments near the mouth of the estuary causing potential increases in water level throughout. Several managed realignment breach sites within the United Kingdom have seen the development of new creek systems with some channel movement and deepening of those existing [18,21]. In the Blyth estuary, located in the east coast of the United Kingdom, the restriction of tidal exchange to small areas leads to increases in flow in the vicinity of a seawall breach, but negligible longer-range effects [31], while Leggett et al. [32] related the increase in estuarine tidal prism owing to realignment to an increase in the estuarine channel cross-sectional area.

It is important to understand the contexts in which these saltmarshes exist, and how these may alter the flow and wave dynamics at sites. For example, while marshes have been demonstrated to be effective in attenuating wave energy along open coastlines that typify marshes on the east coast

of the United Kingdom, it is less well defined for smaller and more sheltered bar built estuaries, which characterize much of the west coast of Wales [3,5,33–35]. These sheltered estuaries are inherently lower energy environments than open coastlines or larger estuaries, which may alter the relationship between saltmarshes and the role of wave dampening for flood mitigation [5,34,36]. Similarly, grazing of saltmarsh vegetation is a popular practice in livestock farming, yet it reduces the vegetation height, and subsequently the resistance to flow and waves. It is hypothesised that allowing grazing of saltmarsh areas can impact the hydrodynamics on marshes, and potentially impair coastal protection function of the marsh [37].

In this paper, we investigate the effects of three fundamentally different flood mitigation intervention strategies on hydrodynamics of the Taf Estuary, located in south-west Wales, United Kingdom, during extreme storm events through a computational modelling study. These flood mitigation interventions are the introduction of a hard defence, the implementation of managed realignment sites, and the controlling of the use of saltmarsh area for grazing by livestock. The Taf Estuary typifies small, sheltered, macro-tidal estuarine systems in Wales, which significantly differ from large estuaries found in the east coast of the United Kingdom. The intervention strategies tested in this study are based on the broad range of flood and coastal erosion risk management measures under consideration by policy makers and stakeholders around the coast of Wales.

## 2. Taf Estuary

The Taf Estuary, located in the Carmarthen Bay of south-west Wales, United Kingdom, is a small funnel-shaped estuary, which covers approximately 8.65 km$^2$ (Figure 1). This is one of the Carmarthen Bay Three Rivers confluences. The estuary is macro-tidal with a mean spring tidal range at the estuary mouth of 7.5 m, a neap tidal range of 3.7 m, and a tidal prism of $17.7 \times 10^6$ m$^3$ [38,39]. The tidal limit of the estuary is located approximately 15 km upstream of its mouth [40]. Two rivers, Taf and Cywyn, feed the estuary. The river Taf brings an average daily freshwater discharge at the head of the estuary of 7.0 m$^3$/s. The extreme high and extreme low discharges have been recorded as 60 m$^3$/s and 0.6 m$^3$/s respectively [41]. No discharge records were found for the River Cywyn, however, the very small catchment area of the River Cywyn suggests a very small freshwater input. The peak tidal currents in the estuary reach 2.2 m/s near the mouth of the estuary [38]. It should be noted that the river flow is very small compared with the large tidal prism owing to the macro-tidal regime.

Long-distance swell waves reach Carmarthen Bay predominantly from the south-westerly direction. Swell wave penetration into the estuary is limited by the orientation of the mouth of the estuary with respect to the wave approach direction [40]. Ginst point and Wharley point are of importance (Figure 1), as they restrict the estuary mouth from the southwest and the northeast. However, locally generated wind waves within Carmarthen Bay can be significant and have a wider array of directions [40].

The Taf Estuary is characterised by typical estuarine features such as sand and mud flats, saltmarshes, and intertidal channels [42]. There are four main areas of saltmarsh (Figure 2), Laugharne Castle, Laugharne South, Laugharne North, and Black Scar, occupying a total area of 279 ha [39]. A single intertidal channel runs through the lower estuary, which then branches into two or three small channels in the upstream. However, these channels evolve very rapidly where more than one channel can be found in the lower estuary from time to time. The channel locations also evolve from time to time.

Historically, the Taf Estuary has been subjected to significant changes prior to the configuration seen at present. Numerous land reclamation projects have been reported in the 17th and 19th centuries, with an estimated 700 ha of drained saltmarsh [39]. In recent decades, the Taf has seen a large increase in saltmarshes, with accretion and progradation on both sides of the estuary, owing to the introduction and spread of *Spartina anglica* species and the movement of Ginst Point [39].

The marshes fringing the Taf Estuary are characterised by the woody shrub *Atriplex portulacoides*, which dominates much of the marsh between the low marsh and mean high water mark. Grasses, such as *Puccinellia maritima* and *Festuca rubra*, as well as the perennial herb *Aster tripolium*, form interspersed patches within the low-high marshes, especially at the marshes around Laugharne and Black Scar

(Figure 2), and extend into the upper marsh above the mean high water throughout most of the estuary. Low marsh areas with adjacent mudflats host dense *Spartina anglica* grass communities, although these are absent in around half of the marshes, where erosion of the banks leads to a sharp transition from *Atriplex portulacoides* to the river channel.

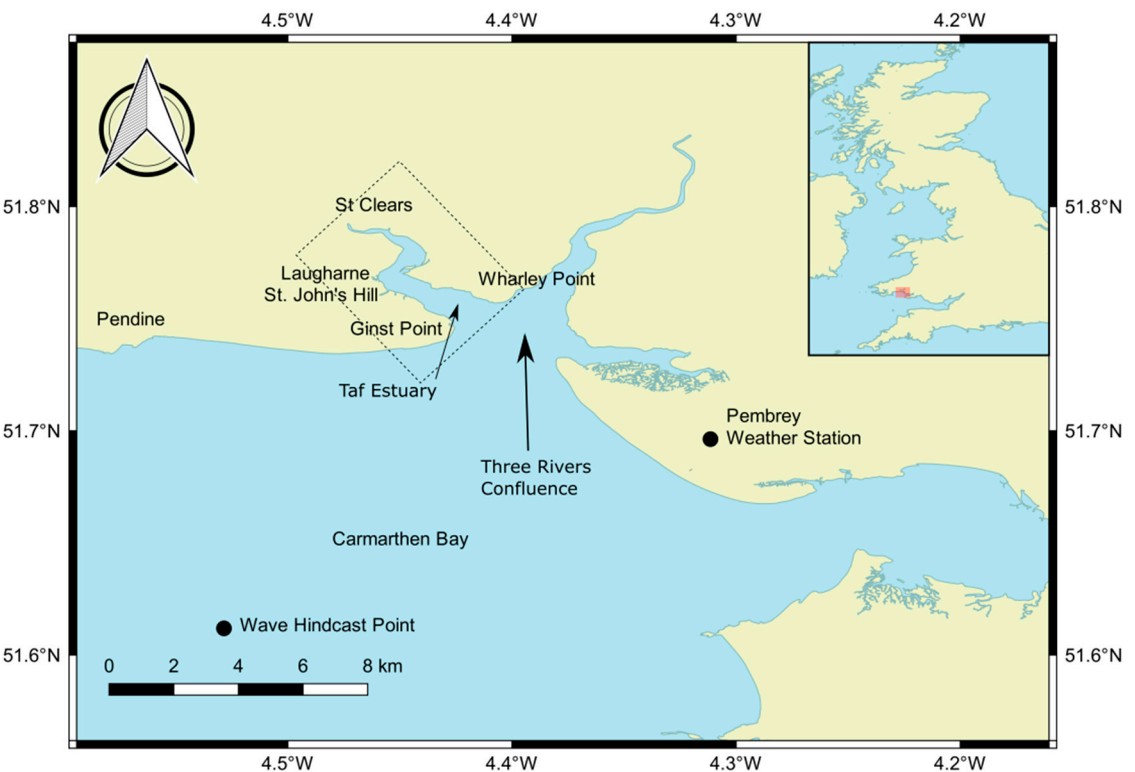

**Figure 1.** Taf Estuary and its location in Carmarthen Bay, south-west Wales, United Kingdom.

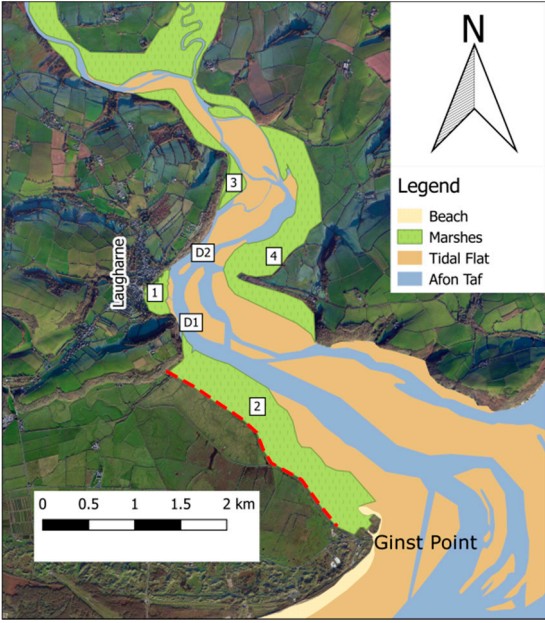

**Figure 2.** Overview of the Taf Estuary and its saltmarshes. Numbers 1, 2, 3, and 4 denote Laugharne Castle, Laugharne South, Laugharne North, and Black Scar marshes, respectively. D1 and D2 highlight the locations of the Acoustic Doppler Current Profiler (ADCP) deployment from 2018 and current meter deployment from 2017, respectively. Dashed red line indicates historic defences.

The Taf represents a typical Welsh estuary in terms of size, tidal characteristics, and morphodynamic features. The historic village Laugharne, located at the fringe of one of the largest marshes of the estuary, which regularly floods during winter storms, attracts the attention of policy makers and coastal managers as there is an urgent need to implement a sustainable flood prevention solution to protect the village. The current management policy in the Taf Estuary is to allow the natural development of undefended shores, and to reduce the risk of flooding and erosion [41]. This results in an array of policy decisions depending on the assets at risk, and the time frame considered. For the eastern bank of the estuary, and the western bank north of Laugharne, no active intervention is chosen. However, to protect the village of Laugharne and further south, a mixture of managed realignment and hold-the-line policies are utilised. Hold-the-line consists of maintaining the current coastline through maintenance or construction of coastal defences to protect against sea level rise and coastal flooding. In the short term, the existing defences constructed for reclamation through the 17th, 18th, and 19th centuries south of Laugharne and between Ginst Point (Figure 2) will be maintained. In the medium to longer term, managed realignment of the section including Laugharne and south to Ginst Point and holding the setback line is considered as an option [41]. This also satisfies the criteria to create a compensatory intertidal habitat, and as this land was previously saltmarsh and reclaimed from the sea, it is ideal for the formation of compensatory saltmarsh. Improvements to reduce flood risk have been undertaken in the village of Laugharne, including individual flood protection and flood warnings to the 43 properties. A proposed surge barrier to protect the village was, however, rejected because of the potential aesthetic impacts, and thus there remains a risk of coastal flooding [41].

The small scale and macro-tidal regime of the Taf Estuary make it very sensitive to any changes in hydrodynamic regime. It has been found that the morphology of the Taf Estuary is highly dynamic, changing rapidly over very small timescales. As a result, any potential change to hydrodynamics of the estuary may have significant implications on the physiology and ecology of the estuary [39,43]. The management interventions considered for investigation for their potential to change the estuarine hydrodynamics are the construction of a hard defence at the village of Laugharne, the implementation of managed realignment, and the use of saltmarsh area for grazing by livestock.

## 3. Methodology

We use computational modelling to investigate the impact of management intervention scenarios on the flood mitigation function of saltmarshes of the Taf Estuary and wider hydrodynamic regime of the Taf Estuary.

### 3.1. Storm Boundary Conditions

#### 3.1.1. Waves

Wave data for the Carmarthen Bay were provided by the Centre for Environment Fisheries & Aquaculture Science (CEFAS), United Kingdom wave hindcast dataset. Wave data from the nearest hindcast output point in Carmarthen Bay (Figure 1) were extracted for the period 1980–2017. The storms occurring during this period were isolated following the method described in Bennett et al. [44], which is based on the storm definition provided by Dissanayake et al. [45]. The threshold storm wave height was taken as 2.5 m, based on the U.K. Channel Coastal Observatory (CCO) guidance [46] for defining storm thresholds. This method provided the peak significant storm wave height, peak period, and the duration of all storms occurring between 1980 and 2017. The Generalised Pareto Distribution (GPD) was then used to determine extreme significant storm wave heights of a range of statistically significant storm conditions from those storm data extracted from the hindcast wave data. The GPD (Equation (1)), which is the combination of three statistical families, was fit to the peak storm wave heights, using the method of Hawkes et al. [47], owing to its successful application on a range of oceanographic variables [48,49]. In Equation (1), $\phi$ and $\xi$ are scale and shape parameters, respectively,

and $u$ is the threshold that ensures model convergence [48]. The R statistical software package ismev [48] was utilised to fit the GPD to the data [50].

$$Pr\{X > x | X > u\} = \begin{cases} 1 + \xi\,\phi^{-1}\,(x - u)^{\frac{-1}{\xi}} & \xi \neq 0 \\ e^{-\frac{x-u}{\phi}} & \xi = 0 \end{cases} \tag{1}$$

The GPD (Figure 3a) is used to determine the significant wave height of storms corresponding to 1 in 1-, 10-, 50-, and 100-year return periods. The average significant storm wave height was taken as the average value from the filtered storm conditions. The maximum storm wave period ($T_{max}$) was determined from the average of the maximum wave period of individual storms extracted from the hindcast wave data. The average of the predominant wave direction of all individual storm events was used as the predominant storm wave direction ($Wd_{avg}$). These conditions are summarised in Table 1.

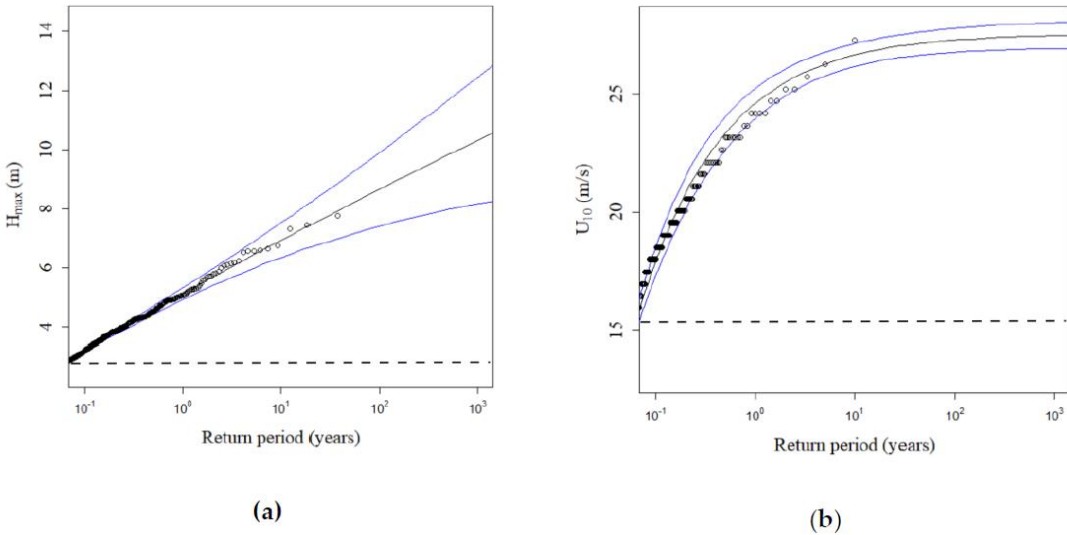

**Figure 3.** Generalised Pareto Distribution (GPD) profiles for wave height in metres (**a**) and wind speed in m/s (**b**). Crosses indicate storm significant wave height and wind speed values, with the GPD fit and 95% confidence intervals indicated by the three curves. Dashed line indicates the threshold level.

**Table 1.** Summary of statistically significant storm wave boundary conditions in Carmarthen Bay. $T_{max}$ is the maximum storm wave period, and $Wd_{avg}$ is the predominant storm wave direction.

| Parameter | Average Storm | 1 in 1 Year | 1 in 10 Year | 1 in 50 Year | 1 in 100 Year | $Wd_{avg}$ (Degrees) | $T_{max}$ (s) |
|---|---|---|---|---|---|---|---|
| **Significant Wave Height (m)** | 3.37 | 5.13 | 6.94 | 8.15 | 8.66 | | |
| **Wind Speed (m/s)** | 16.75 | 24.59 | 26.64 | 27.15 | 27.27 | 225 | 6.82 |
| **Easterly Wind Speed (m/s)** | 12.21 | 13.21 | 15.06 | 16.28 | 16.28 | | |

To determine corresponding wind characteristics during storms, hourly wind outputs from the nearby Pembrey weather station (Figure 1) for the period of 2002–2009 were used. Similar to the analysis of storm wave data, the GPD was fitted to the wind data. The predominant wind direction was determined from the observed wind data during storm conditions. The return level plot for the storm wind velocity is shown in Figure 3 (right), while wind speeds with selected return periods are summarised in Table 1. Easterly storm wind conditions were determined by filtering wind approaching from the east (between 45 and 135 degrees), before fitting the GPD to the data.

The storm wave and wind conditions given in Table 1 were then used to generate time-varying wind and storm profiles using a three-point spline curve, which closely represents observed storm profiles. In this, the storm begins when the incident wave height exceeds the pre-selected threshold

wave height and ceases when the wave height becomes smaller than the threshold. The storm peaks halfway between the beginning and the end of the storm, thus creating a symmetric storm profile. The corresponding wind conditions for the chosen storm wave return period follow the same three-point spline shape. The storm duration was defined based on the length of the storm surge profile for the Mumbles tide gauge, provided from the dataset of McMillan et al. [51], utilised to determine storm water levels. This provided a storm duration of 75 hours for the Taf Estuary.

### 3.1.2. Water Levels

Statistically significant water level variations during storms are determined following the method given in McMillan et al. [51]. Using sea water level data supplied by the National Tide and Sea Level Facility of the UK (NTSLF), they had determined peak water levels and storm water level profiles using the skew surge joint probability method (SSJPM) for forty U.K. national network (class A) tide gauge sites. Their analysis provides sea levels with a range of return periods at 2 km spacing around the U.K. coastline.

The total water level during extreme events is a combination of the astronomical tide and the storm surge. Scaled surge shape (Figure 4) for Mumbles tide gauge (the closest tide gauge to Carmarthen Bay) can be determined following the method used in their analysis. This allows the derivation of appropriate time-varying total water level for this location. The guidance provided by McMillan et al. [51] suggests that the base astronomical curve should be halfway between the mean high water spring tide (MHWS) and highest astronomical tide (HAT), which, in this case, is 4.35 m ordnance datum (OD). This was also utilised as the average storm water level for the analysis. To generate the final time-varying water level during the desired range of storm conditions, the peak storm water level was combined with the time-varying surge profile for each return period to scale up the base astronomical curve (Figure 4) to accommodate storm surge. The peak storm water level conditions are summarised in Table 2. In this analysis, we assume that the storm peak coincides with high tide and the maximum surge occurs at the peak of the storm to represent the worst-case extreme event scenario.

**Table 2.** Peak storm water levels in Carmarthen Bay, determined following McMillan et al. [51].

| Parameter | Average Storm | 1 in 1 Year | 1 in 10 Year | 1 in 50 Year | 1 in 100 Year |
|---|---|---|---|---|---|
| **Peak Storm Water Level (m)** | 4.35 | 5 | 5.26 | 5.43 | 5.51 |

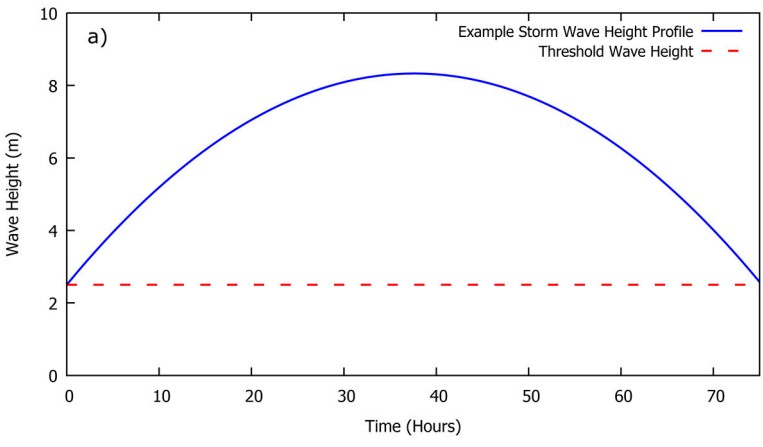

**Figure 4.** *Cont.*

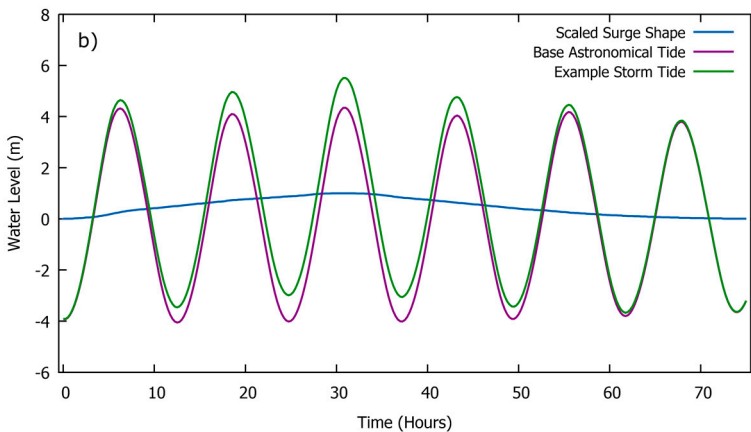

**Figure 4.** An example the (**a**) storm profile and (**b**) surge profile [51] of the base astronomical tide and storm tide at Carmarthen Bay.

### 3.1.3. Storm Conditions

The storm wave, wind, and water levels generated in Section 3.1 are used to generate two extreme storm conditions to be used in this paper: (i) a south westerly storm, which is characterised by 1 in 50-year return period peak storm wave height, wind speed, and water level; and (ii) an easterly storm, which combines 1 in 50-year wind speed with average storm wave height and water level (Table 3). Average storm wave height and water level are used because extremely easterly wind does not coincide with extreme wave or water level conditions, which propagate from the Atlantic Ocean.

**Table 3.** Summary of storm boundary conditions.

| Storm Condition | Peak Storm Water Level (m) | Peak Storm Wave Height (m) | Peak Storm Wind Speed (m/s) | Predominant Wind Direction (Degrees) |
|---|---|---|---|---|
| South Westerly Storm | 5.43 | 8.15 | 27.15 | 225 |
| Easterly Storm | 4.35 | 3.37 | 16.28 | 90 |

In this study, we combine extreme storm wave, water level, and wind conditions with maximum river discharge to investigate the impacts of saltmarsh management interventions under the worst-case scenarios. It should be noted that the river discharge of the Taf Estuary is very small compared with the tidal flows resulting from the macro-tidal regime. Therefore, the contribution of river discharge into the estuarine hydrodynamics is small.

### 3.2. Modelling Approach

The computational coastal modelling suite Delft3D [52] was used to build a computational model of the Taf Estuary to investigate changes in estuary hydrodynamics owing to a range of coastal saltmarsh management intervention scenarios. Delft3D simulates waves, hydrodynamics, sediment transport, morphological evolution, ecology, and water quality of coastal and estuarine systems and has been extensively used worldwide. The model has been recently used successfully to investigate the ecology–hydrodynamic interface in a variety of estuarine systems with different saltmarsh species and hydrodynamic conditions (e.g., [53–55]). It allows for the implementation of various saltmarsh features such as vegetation characteristics and different sediment types and allows implementation of hard defences. For this study, the 2D depth averaged version of Delft3D FLOW is utilised, solving the unsteady shallow water equations with the hydrostatic pressure assumption.

The Taf Estuary model will provide waves and hydrodynamics of the estuary and capture the impacts of river flow, saltmarsh ecology, and wave–current interactions on estuarine hydrodynamics. The Taf hydrodynamic model uses curvilinear grids created using the RGFGRID facility in Delft3D for the entire model domain. Encompassing the majority of the Taf Estuary, the model domain extends

out into the Carmarthen Bay to a depth of 22 m (Figure 5b), to avoid potential boundary effects on the estuarine hydrodynamics. Three computational domains are combined using the 'domain decomposition' method in order to allow local grid refinement to capture greater details within the Taf, and near the village of Laugharne, which has been flooding frequently. Grid cells gradually refine from approximately 200 m × 400 m at the offshore boundary to 10 m × 10 m over the Laugharne Castle marsh (Figure 5a).

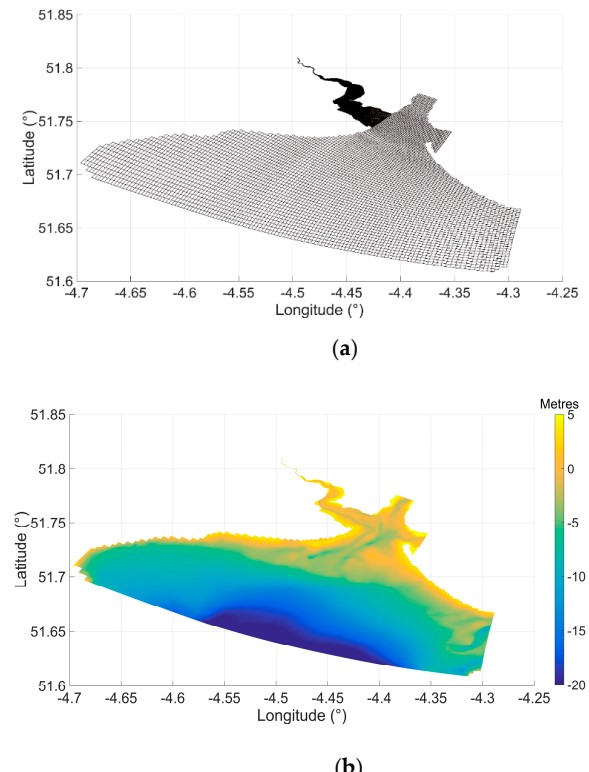

**Figure 5.** Taf Estuary model domain. (**a**) Model grids cover the Taf Estuary and Three River confluence, extending into Carmarthen Bay. (**b**) Model bathymetry. Colour bar in metres ordnance datum (OD).

Existing bathymetric data of the Taf Estuary were poor, largely because of available LiDAR data being out of date. Therefore, it was decided to carry out new, high resolution bathymetric surveys to capture recent channel movement within the estuary. Bathymetric measurements were done using a small rigid-hull inflatable boat equipped with a high-resolution single-beam and sidescan sonar system (Simrad NSS Evo3 with Lowrance Structurescan 3D +Hi/Lo CHiRP Transducer) during spring high tides between 21st and 25th of August 2017. The bathymetry of Carmarthen Bay was obtained from the U.K. Hydrographic Office (UKHO), who surveyed the bay with a multi-beam echo-sounder at 2 m (X-Y) resolution in 2013. This dataset extends up to the 5 m depth contour in Carmarthen Bay. Finally, the bathymetry of the Three Rivers confluence, which connects the Taf Estuary to Carmarthen Bay's 5 m depth contour, was obtained from Admiralty charts from 1977. While it is significantly older than any other source used, the channel positions provide a good qualitative comparison with recent satellite imagery (e.g., Google Earth). In the absence of a complete bathymetry dataset for Taf Estuary and the surroundings, it was necessary to combine the three datasets measured at different times to provide the model bathymetry. This impact is investigated when validating the model against hydrodynamic observations within the Taf.

### 3.3. Vegetation Module

Saltmarsh vegetation is modelled with plant geometry simplified as rigid cylinders that are parameterized by plant height $h_v$, stem diameter $b_v$, and plant density $n_v$ [56]. The stem diameter

reduces in the upward stem direction. The validity of these assumption builds on the woody stems of *Atriplex portulacoides* and its successful application in prior numerical modelling studies, including saltmarshes (e.g., Ashall et al. [57]).

The drag force induced by rigid cylindrical saltmarsh vegetation on currents can be modelled as a sink term in the momentum equations, according to Equation (2) [58].

$$\vec{F} = \frac{1}{2}\rho C_D b_v n_v |\vec{u}| \vec{u}$$ (2)

where $F$ is the drag force produced by vegetation per unit volume in N/m$^3$, $\rho$ is the density of the fluid in kg/m$^3$, $C_D$ is a dimensionless drag coefficient, and $u$ is the local flow velocity in m/s.

Saltmarsh vegetation dampens waves owing to the work done by the drag force (Equation (2)) on plants stems [56], implemented in Delft3D WAVE via Equation (3).

$$\frac{\partial E c_g}{\partial x} = \langle \int_{-h}^{-h+h_v} \vec{F}\vec{u}\,dz \rangle$$ (3)

where $E = \frac{1}{8}\rho g H^2$ is the wave energy density in N/m and $c_g$ is the group velocity in m/s. Furthermore, $h$ is the water depth in m and $h_v$ is the vegetation height in m. The horizontal axis $x$ is in the direction of wave propagation and $z$ is the vertical axis in the water column. The brackets ($\langle \rangle$) denote phase-averaging.

Importantly, the drag coefficient $C_D$ is the only parameter that cannot be measured in the field a priori. Therefore, the value $C_D = 1$ is selected based on experimental studies with stiff cylinders in unidirectional flow [59] and waves based on conditions in the Taf Estuary [60]. $C_D = 1$ was also successfully applied in recent modelling studies [54,57].

The vegetation parameters are selected based on the samples of *Atriplex portulacoides*, as Delft3D WAVE can only account for a single species. *Atriplex portulacoides* is a stiff woody shrub with 53% cover of saltmarsh platforms in the Taf Estuary. It is most dominant in the lower and middle marshes that are most often inundated. Vegetation density was measured by the authors using 125 haphazardly selected 25 × 25 cm quadrants from the Laugharne marsh in 2017. Furthermore, plant height and stem diameter were measured using 72 samples at the Biosciences Laboratory of Swansea University, UK. It was found that the mean stem density is 2275 stems/m$^2$, mean stem height is 34 mm, and stem diameter is 3.3 mm at the substrate and 1.86 mm at the tip of the plant. These values are representative of vegetation found in the marshes of most Welsh estuaries. The vertical variation in stem diameter was explicitly accounted for in Delft3D FLOW, but an average value of 2.58 mm was used in Delft3D WAVE.

Finally, the river discharge is set to represent the most extreme conditions observed in the Taf, providing the worst-case storm scenario. Specifically, it has been set at a constant discharge of 60 m$^3$/s, which is the highest measured discharge [38]. The full list of model parameters used in Delft3D for the Taf Estuary is provided in Table 4.

**Table 4.** Model input parameters.

| Parameter | Symbol | Value | Unit | Motivation |
|---|---|---|---|---|
| Plant Width | $b_v$ | 2.58 * | mm | From field campaign |
| Plant Height | $h_v$ | 34 | mm | From field campaign |
| Plant Density | $n_v$ | 2275 | $m^{-2}$ | From field campaign |
| Drag Coefficient | $C_D$ | 1.0 | - | Tanino and Nepf [56]; van Veelen et al [57] |
| Bed roughness | $C_b$ | 65 | $m^{1/2}/s$ | Marciano et al [61] |
| River discharge | $Q$ | 60 | $m^3/s$ | Ishak [38] |
| Water density | $\rho_0$ | 1025 | $kg/m^3$ | Well-mixed estuary |
| Horizontal eddy viscosity | $K$ | 1 | $m^2/s$ | Mariotti and Canestrelli [62] |

* On average. In fact, the plant width is a function of height. The width is 3.3 mm at the substrate and 1.86 mm at the tip.

### 3.4. Model Validation

#### 3.4.1. Validation of Water Depth

Water depth data obtained from an ADCP deployed in the Taf was used to validate the numerical model. The ADCP was deployed during 10 tidal cycles between 10 and 16 June 2018 in the main channel next to St John's Hill (see Figure 2, D1). Water depths were measured in bins of 50 cm, starting from a minimum of 1 m water depth. The model was run for the same period with tidal forcing only.

Comparison between the model results and measured water depths (Figure 6A) at this location shows excellent agreement in phase and good agreement in amplitude. The visual difference in peak high-water level between the measurements and model relates to the bin classifications in the measured data set. The values plotted are the minimum value of each bin and the actual value may be up to 50 cm higher. This is particularly pronounced during the peak high waters, where a successive bin may not be reached.

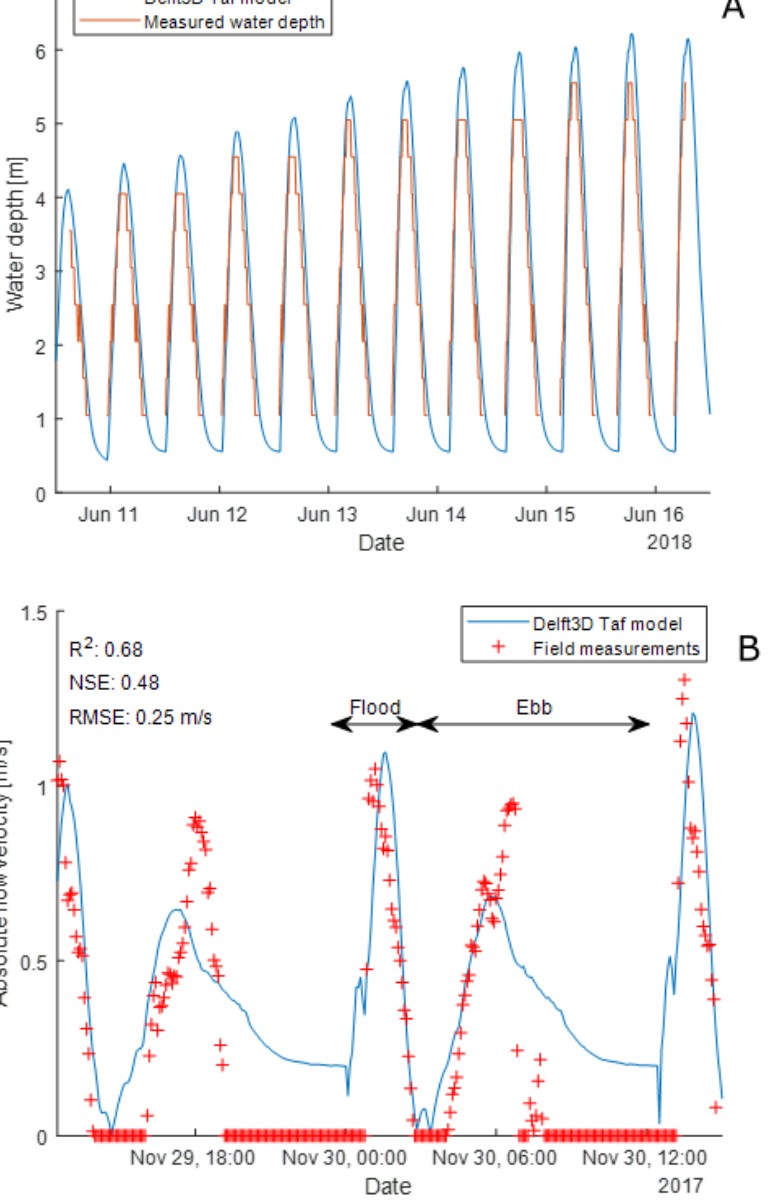

**Figure 6.** *Cont.*

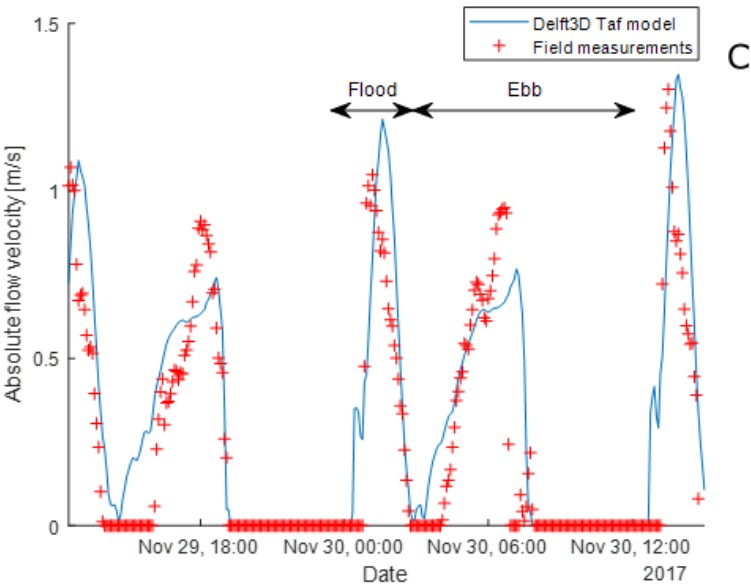

**Figure 6.** Taf Estuary computational model validation. (**A**) Comparison of modelled water depth with field measurements in the main estuary channel, next to St. John's Hill. (**B**) Comparison of modelled flow velocity with field measurements in the main estuary channel, at the exact location where the velocity meter was anchored. (**C**) Same as B, but at a location on the Laugharne saltmarsh, in the proximity to the channel. In all plots, the blue line depicts modelled outputs and the red line/red crosses depict measured values. NSE, Nash–Sutcliffe efficiency coefficient; RMSE, root mean square error.

### 3.4.2. Validation of Flow Velocity

A single point current meter was deployed in the main channel upstream of Laugharne Castle marsh (see Figure 2, D2) between 13:00 on 28 November 2017 and 14:45 pm on 30 November 2017. Comparison of modelled flow velocities at the location of deployment with modelled results shows reasonable agreement (Figure 6B,C). Coefficient of correlation $r^2 = 0.68$, Nash–Sutcliffe efficiency coefficient [63] $NSE = 0.48$, and root mean square error $RMSE = 0.25$ m/s indicate that the model predicts flow velocities adequately for an estuary with significant tidal asymmetry. In particular, the flood dominance of the estuary is well reproduced by the model. Furthermore, the phase and amplitude of the flood tidal current match well with the measurements. The initial stage of the ebb tidal currents is also modelled correctly. However, measured ebb currents have a stronger maximum velocity and the ebb tide is of shorter duration. This could be the result of grounding of the meter on one of the tidal flats next to the channel. As the meter was fixed to a rope, it was free to move around the anchor point and could have travelled onto the flats. This is supported by the model results of a grid point in close proximity, but on a tidal flat. Unlike the main channel, the flat surfaces will no longer contribute to drainage of the estuary. Indeed, Figure 6C shows that the ebb phase corresponds much better with measurements for such a location. The validation results highlight that the combination of datasets to provide the model bathymetry provides a reasonably accurate representation of the current estuarine configuration.

### 3.5. Implementation of Coastal Management Interventions

Some places surrounding the Taf Estuary are subjected to repeated flooding events, which has significant implications on the socioeconomics of nearby coastal populations. Therefore, some flood mitigation interventions surrounding saltmarshes have been proposed. Those interventions reflect both "hard" and "soft" engineering options, as well as changes in land use and environment.

### 3.5.1. Managed Realignment

The Countryside Council for Wales Report of Cousins et al. [64] suggested two potential areas for managed realignment in the Taf Estuary (Figure 7). Mwche farm and Mylett farm are currently privately-owned agricultural land. Together, they provide 77.71 ha of land for potential realignment through breaching of existing defences. As former saltmarsh areas, these sites have the potential to revert back into saltmarshes [65].

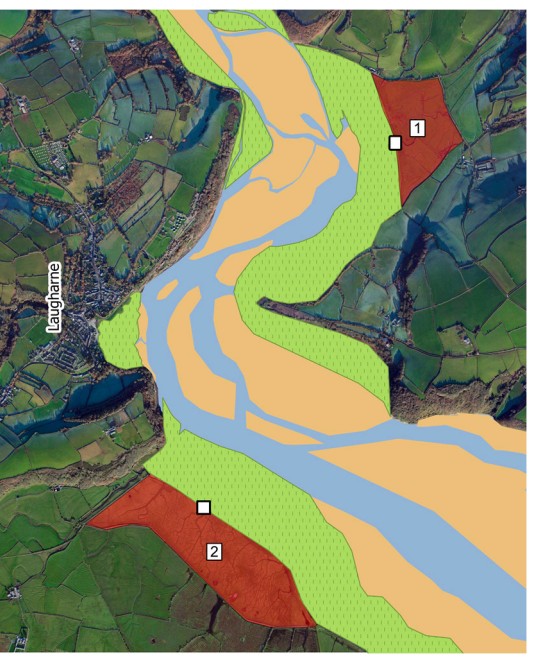

**Figure 7.** Proposed Taf Estuary managed realignment sites indicated by Cousins et al. [64]. Numbers 1 and 2 indicate Mwche and Mylett farm sites, respectively. White squares indicate suggested breach locations.

To implement managed realignment within the model, several measures were taken. Firstly, the model domain was extended to include the two sites. Then, following the advice of Bristow and Pile [39], and commonplace for managed realignment sites within the United Kingdom, the existing flood defences were artificially breached. The breach consists of the removal of a section of the defence in the alongshore direction to allow for water to flow into the site. It is important for the size of the breach to be designed such that it does not lead to undesired morphological effects owing to significant flow velocities through the breach. When breaches are first created, if they are not significantly wider than the projected regime width of the ebb channel, excessive ebb currents may cause scour and changes to the position of the main channel of the estuary [2]. Equation (4) was used to determine the breach width as a function of tidal prism at each location [32]. In Equation (4), $W$ = breach width alongshore and $TP$ = MHWS tidal prism. The fit parameters $\alpha$ and $\beta$ were estimated as 37.9 and $1.8 \times 10^{-6}$, respectively, based on previous observations of historical breaches at 23 sites that had been enclosed for agricultural use.

$$W = \alpha e^{\beta TP} \tag{4}$$

Cousins et al. [64] provide values for the increases in tidal prism as a result of the demolition of existing flood defences at the Mylett and Mwche farm sites as 107,750 m$^3$ and 46,350 m$^3$, respectively. Equation (4) then gives alongshore breach widths of 46 m and 41 m for the two sites, respectively. The breaches at both sites were placed at the end of existing creek networks, and near the locations of relic creeks from the reclaimed saltmarsh. These creek systems were then connected through lowering of the topography on either side of the breach.

### 3.5.2. Grazing

Grazing of saltmarshes by local livestock, which is a common phenomenon in most Welsh estuaries, reduces the height of saltmarsh vegetation. This can have potential benefits for biodiversity [66] and increase plant stem density [37], which can contribute to diminishing flood propagation over the marsh over the long term. It may, however, reduce the vegetation height, and subsequently the resistance to wave and flow propagation on the marsh, thus diminishing their coastal defence function [37] over short-term timescales. To investigate the wider impacts of saltmarsh grazing in the Taf Estuary, the extreme case in which all saltmarsh areas highlighted in Figure 2 are grazed was investigated. Grazing was introduced into the model by artificially removing the vegetation cover from the marshes, which replicates the extreme scenario.

### 3.5.3. Hard Defences

To protect the village of Laugharne from frequent flooding in the future, construction of a surge barrier has been debated. Although it has long since been rejected by the local communities because of the fears about the aesthetic impact and restricted access to the marsh as a recreation ground of such a measure [41], some form of hard defence may be necessary if flooding intensifies in the future. To investigate the potential impacts that a surge barrier may have within the estuary, a thin dam was implemented within the model at the boundary of Laugharne (Figure 8). This prevents any flow from entering the village, eliminating any flooding, if the barrier is high enough to resist rising water levels and wave propagation during a storm.

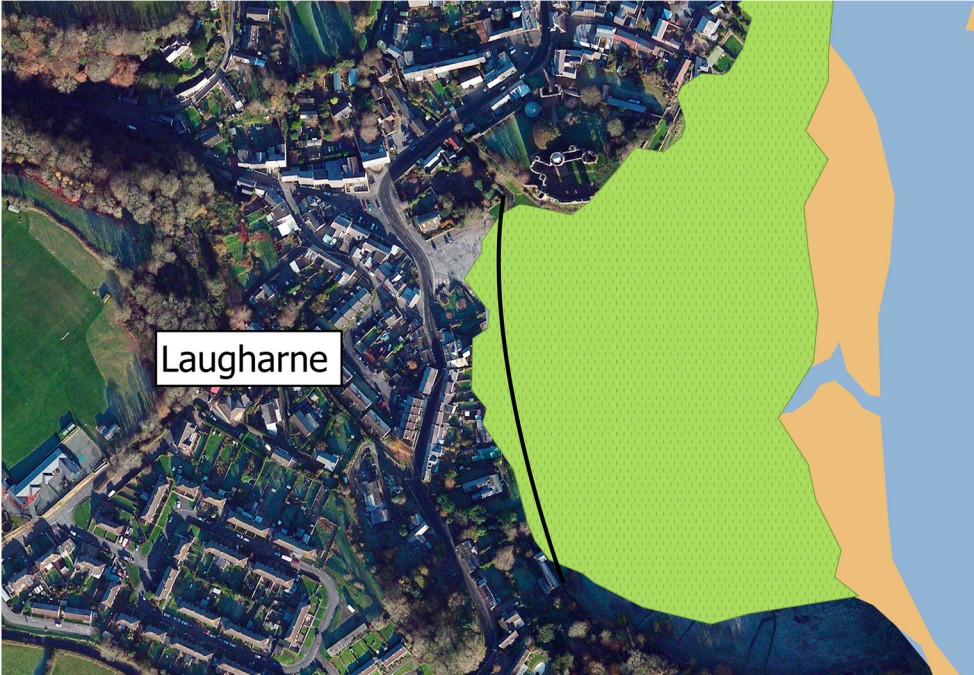

**Figure 8.** Position of the hard flood defence (shown in black line) protecting the village of Laugharne.

## 4. Results

The hydrodynamics of the Taf Estuary, after the implementation of each flood defence intervention scenario mentioned in Section 3.5 during the two selected storm conditions, were simulated. The current hydrodynamic state of the estuary is taken as the baseline scenario to compare and contrast the impacts of those interventions on the overall hydrodynamic regime. It is useful to investigate the impact of management interventions on the hydrodynamics of the estuary as it may have linkages to sediment transport and morphodynamic evolution including saltmarsh erosion and intertidal channel migration.

### 4.1. South-Westerly Storm Conditions

The storm water level in the Taf Estuary and the water level differences between the selected management intervention scenarios and 'baseline' scenario during the peak of the south-westerly storm are shown in Figure 9. In the current undisturbed state of the Taf Estuary, tidal extent during the peak of the storm reaches the landward edges of the marshes, with water depths in the range of 1–2 m. With the introduction of a hard defence to protect the village of Laugharne, little change in the peak storm water level is observed. Behind the structure, there is no water, as expected, but beyond this, there are no noticeable changes. The change in water level owing to marsh grazing is also insignificant. While the water level difference between undisturbed and managed realigned sites is obviously large, there is no noticeable difference seen outside of the marsh areas of the estuary.

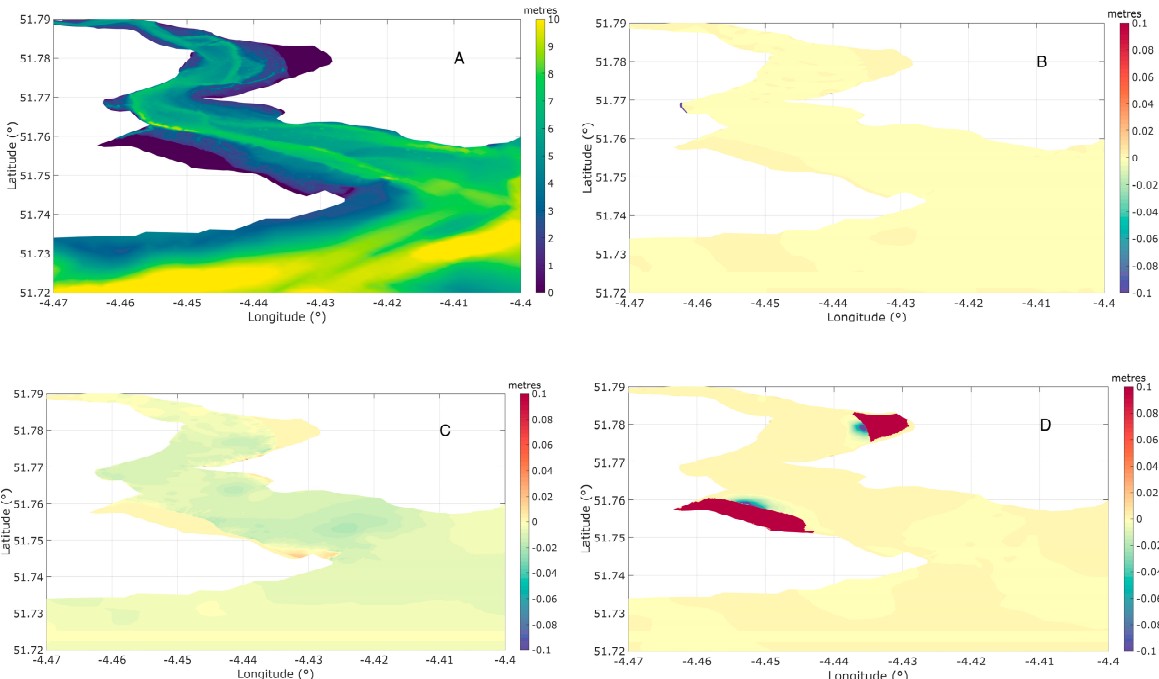

**Figure 9.** Comparison of peak storm water depths at the Taf Estuary under the selected intervention scenarios during the 1:50 year south-westerly storm. (**A**) Current condition, (**B**) difference between defended and current condition, (**C**) difference between grazed and current condition, and (**D**) difference between managed realignment and current condition.

A comparison of change in wave height in the estuary as a result of interventions is shown in Figure 10. In the current situation, the largest wave heights are seen in the areas around the mouth of the estuary (1.2–1.4 m) (Figure 10). At the edges of the marshes, close to the tidal channel location, wave heights reach 0.6–0.7 m, while those on the marshes reduce to less than 0.2 m (Figure 10A). It should be noted here that, during westerly storms, local waves are generated within the estuary owing to strong westerly winds and, therefore, the wave climate in the estuary is complex. The hard defence at the landward edge of Laugharne marsh causes very small changes in wave height coincident with the channel at the seaward boundary of the marsh, with a maximum difference of 0.02 m. With the reduction in wave attenuation owing to the lack of vegetation due to grazing, the wave heights within the estuary are generally increased. Wave height changes on marsh areas are noticeable with a maximum increase of 0.1 m; however, within the channel system, reasonable increases are seen (~0.02–0.05 m). Similar to the grazed case, managed realignment causes a general increase in wave height throughout the estuary. Except when intervened with hard defences, the other two intervention

scenarios increased the wave height on the marsh by around 0.1 m, which may be significant in terms of flooding and marsh erosion.

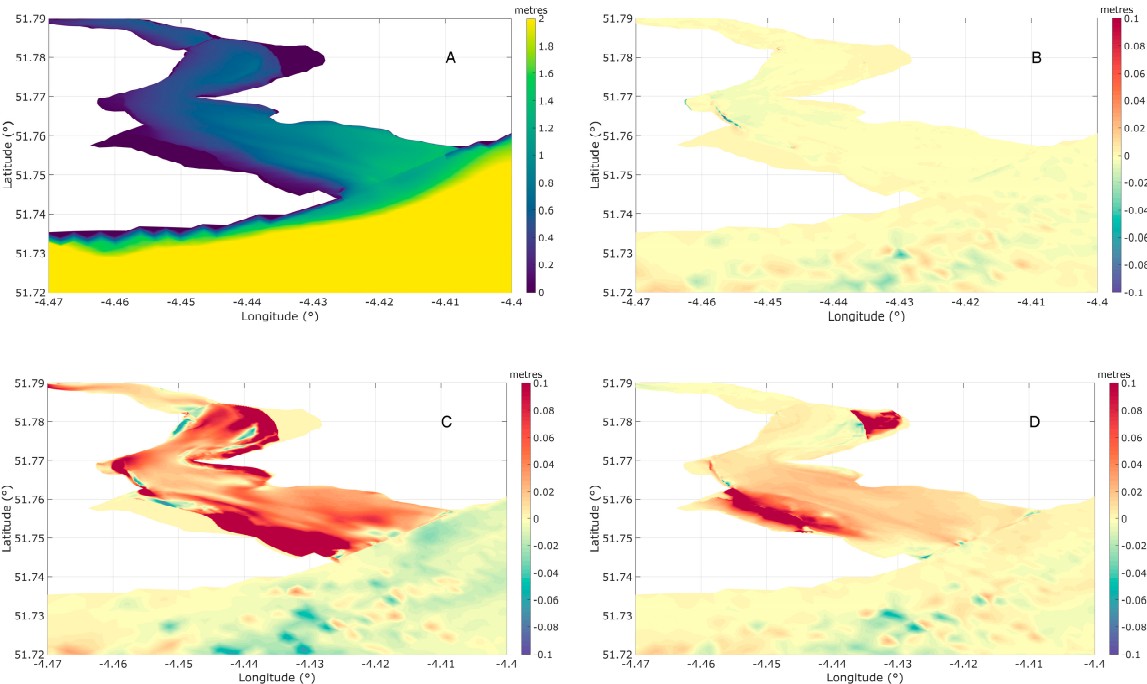

**Figure 10.** Comparison of peak storm wave height in the Taf Estuary during the 1:50 year south-westerly storm. (**A**) Current condition, (**B**) difference between defended and current condition, (**C**) difference between grazed and current condition, and (**D**) difference between managed realignment and current condition.

Using a transect running along the intertidal channel from the mouth of the estuary to Laugharne Castle marsh (T4, Figure 11), the variability of peak storm wave height along the estuary under the coastal management interventions is compared in Figure 12. The variation in wave height along the estuary is clear, with the largest wave heights (~1.3 m) near the mouth of the estuary, which reduce along the transect further into the estuary. There are slight differences throughout the whole estuary, with the largest differences being closer to the Laugharne South and Laugharne Castle Marshes, as the estuary becomes more sheltered.

During the south-westerly storm, currents in the Carmarthen Bay are higher than those in the estuary (Figure 13A), where strong (>1 m/s) alongshore currents propagate along Pendine towards Ginst point. Within the estuary itself, largest flow velocities (~0.5 m/s) are limited to the two main channels and the connecting creek systems, which feed water into the saltmarshes. Flow velocities on the marshes are much smaller (<0.1 m/s) than that in the estuary. The introduction of hard defences did not show a significant change in flow velocities in the estuary or on the marshes. The impact of grazing on flow velocities on the marshes is considerable. Currents across all marsh areas are increased by a maximum of 0.5 m/s, where changes are seen across the majority of Laugharne South, Laugharne North, and Black Scar marshes. In marsh areas close to the main tidal channel, and further north, reductions in current magnitude are observed (~−0.2 m/s). For the managed realignment case, differences are limited, although a significant increase in current magnitude (~0.5 m/s) is observed near the breach, extending adjacent to the wall at both sites, while inside the sites, increases are smaller in magnitude (~0.1 m/s).

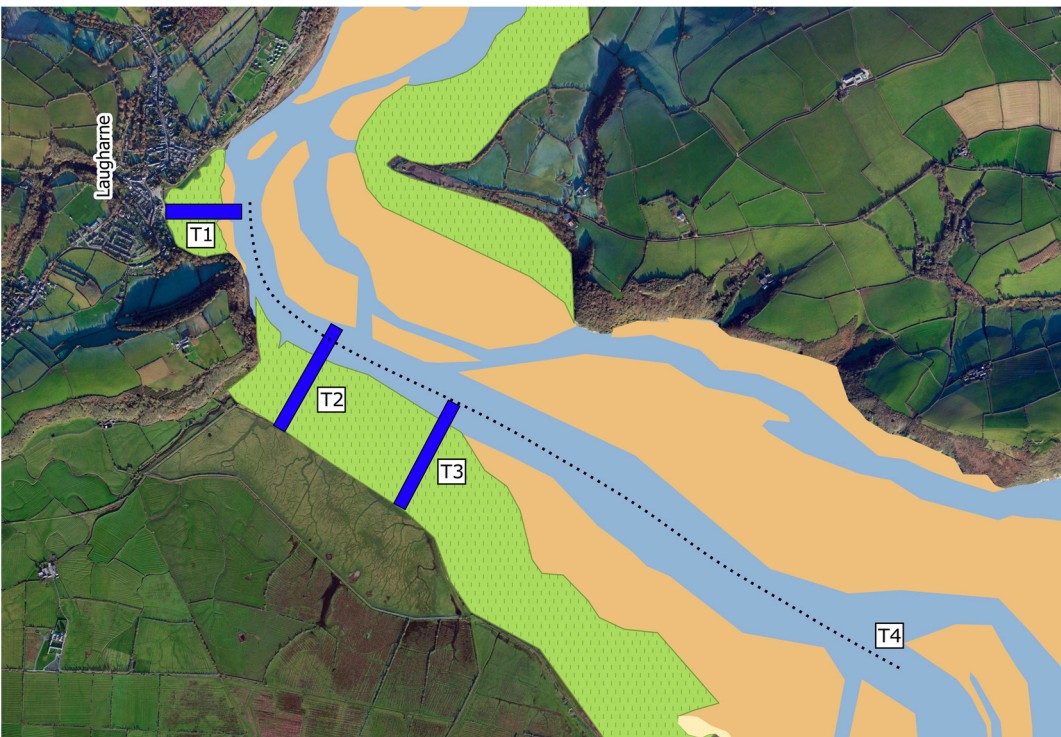

**Figure 11.** Transects and point locations selected for detailed hydrodynamic analysis. T1–4 and P1–4 represent the locations of transects 1–4, respectively.

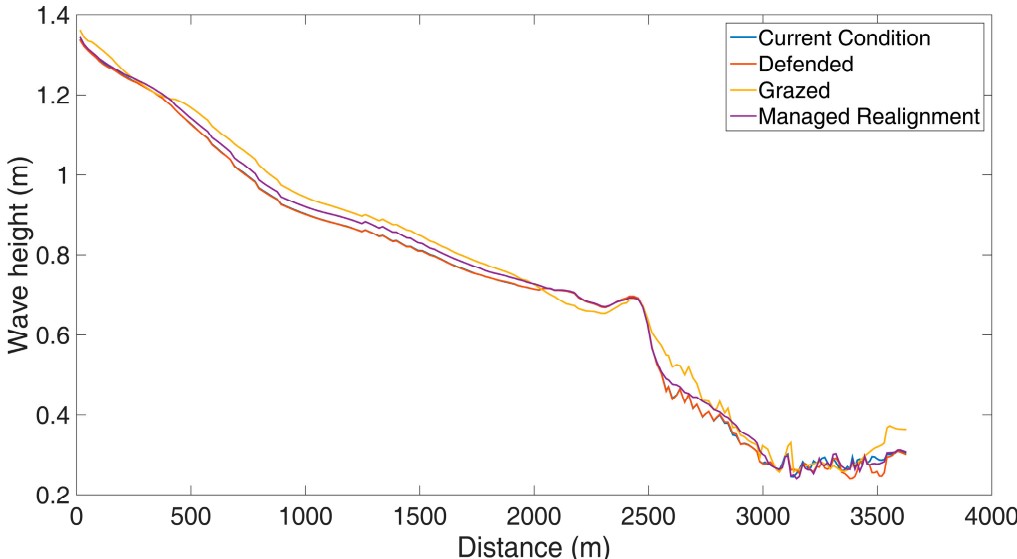

**Figure 12.** Comparison of peak storm wave height along T4 during the 1:50 year south-westerly storm. Distance is measured from the mouth of the estuary in Figure 11.

Peak ebb currents for the south-westerly storm condition (Figure 14A) show a similar pattern to the peak flood currents (Figure 13A), with the strongest (>1 m/s) alongshore currents in the bay propagating from the estuary mouth. Inside the estuary, the largest flow velocities (~0.8 m/s) occur as the estuary narrows between Laugharne Castle and Black Scar marshes, with large velocities (~0.5 m/s) in the main estuary channels. As with the flood currents, peak ebb flow velocities on the marshes are much smaller (<0.1 m/s) than those in the main channels and outside the estuary. While hard defences again showed no impact, the effect of grazing on flow velocities on the marshes is substantial. All marsh areas again showed increased flow velocities, with the largest differences (~0.5 m/s) occurring on Laugharne South marsh near the mouth of the estuary. The main tidal channel, from the mouth of the estuary to Laugharne Castle marsh, and further north, shows a reduction in current magnitude (~−0.2 m/s). The differences due to the managed realignment case are again limited, although there are increases in current magnitude (~0.3 m/s) near both breach sites, extending adjacent to the wall as water leaves the realigned areas.

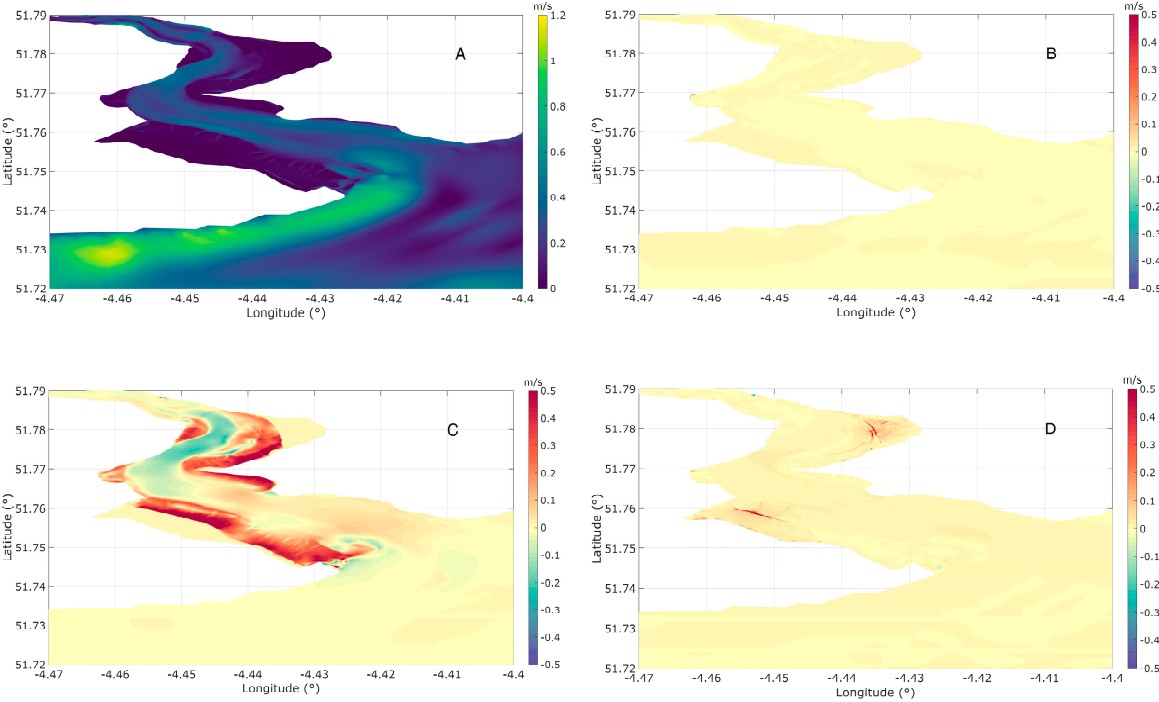

**Figure 13.** Comparison of peak flood current magnitude during the 1:50 year south-westerly storm. (**A**) Current condition, (**B**) difference between defended and current condition, (**C**) difference between grazed and current condition, and (**D**) difference between managed realignment and current condition.

Although the above results show snapshots of the implications of different management interventions on Taf Estuary of its water level, some notable changes to waves and currents were seen, which may have significant implications on marsh areas. Therefore, three saltmarsh transects across the marsh areas (T1, T2, and T3) were chosen to directly compare the differences of hydrodynamics between the selected scenarios (Figure 11).

Wave height variation across the three selected transects is shown in Figure 15. The conditions at T1, located at Laugharne marsh (Figure 11), are summarised in Figure 15A. There are only minor differences in wave height between the current and defended conditions at this location. However, for the grazed case, wave heights are larger than that of the undisturbed scenario across the majority of the transects, with large differences (~0.2 m) between 100 and 250 m, with an increase of 0.05 m towards the mouth of the estuary. The differences seen between the current condition and managed realignment case at transect 1 occur in the channel with a slight increase in the wave heights between 150 and 175 m.

At transect 2, located at Laugharne South marsh (Figure 11), wave height does not change as a result of coastal defences. The impact of grazing causes an increase in wave heights of a maximum of 0.1 m between 150 and 375 m, and a rapid decrease afterwards to the current situation. The impact of managed realignment is clearer with an increase in wave height across almost across the entire transect, with a maximum increase from the current situation around 0.15 m occurring 300–450 m across the marsh. This is potentially owing to waves generated within the realignment site propagating into the estuary through the breach. With the orientation of the predominant wind direction with respect to the estuary, waves propagate offshore of Laugharne South marsh under south-westerly storm conditions. Wave heights at transect 3 increased as a result of grazing and managed realignment. However, wave height stays the same with hard defences. The maximum increase in wave height as a result of grazing and managed occurred between 250 m and 320 m, owing to the growth of waves across the marsh platform in the absence of vegetation, and owing to waves generated, and propagating from, inside the realignment site.

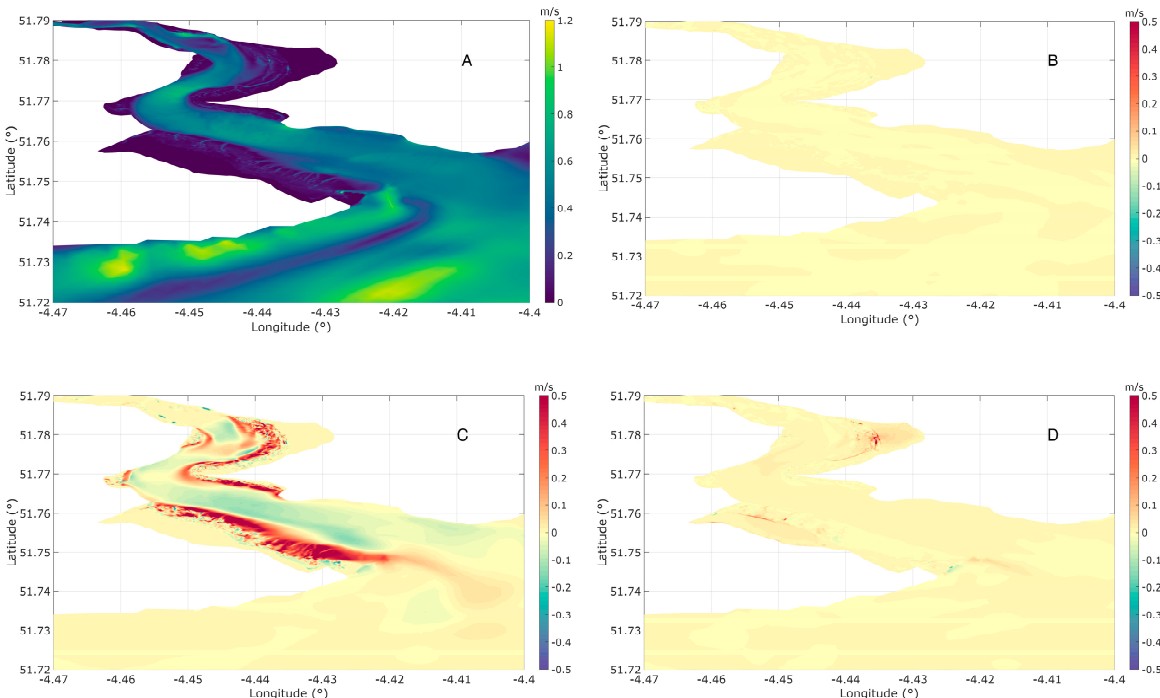

**Figure 14.** Comparison of peak ebb current magnitude during the 1:50 year south-westerly storm. (**A**) Current condition, (**B**) difference between defended and current condition, (**C**) difference between grazed and current condition, and (**D**) difference between managed realignment and current condition.

It should be noted that, in the case of south-westerly storms, the predominant wind direction is also from the south-west. Therefore, most waves on the marsh may be locally generated.

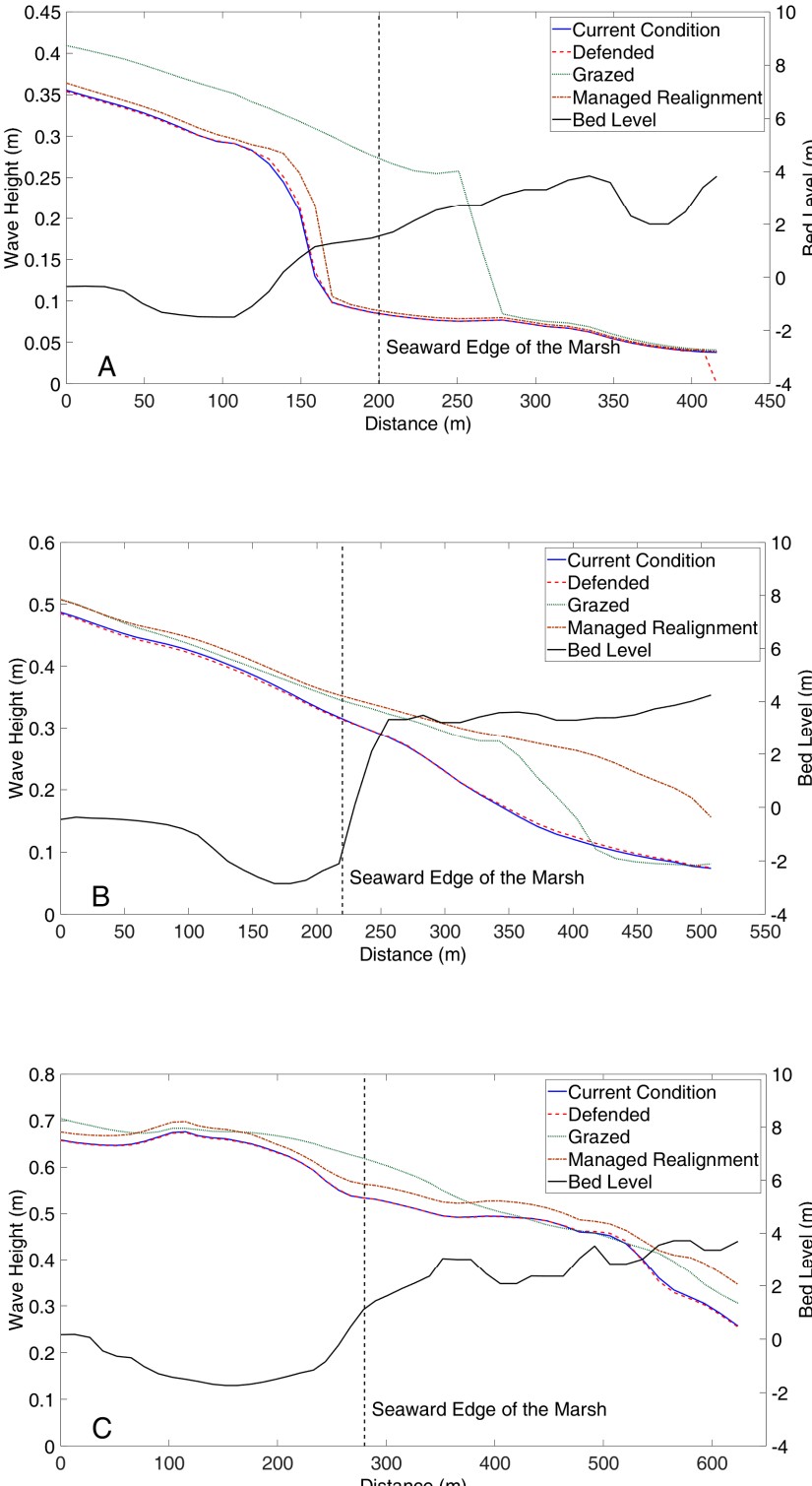

**Figure 15.** Comparison of peak storm wave height for the 1:50 year south-westerly storm condition at T1 (**A**), T2 (**B**), and T3 (**C**). Initial bed level, given in black line, is in m OD. Right hand side is the landward boundary.

### 4.2. Easterly Storm Conditions

Figure 16 shows the spatial distribution of peak storm wave heights in the Taf Estuary for the 1:50 year return period easterly storm. Waves as high as 1.5 m can be seen at the mouth of the estuary. Compared with Figure 10 for the westerly storm with the same return period, owing to the propagation of wind from the east, locally generated waves propagate further onto Laugharne South and Castle marshes (maximum wave height of around 0.8 m at the seaward edge of the marshes), while north of Laugharne, wave heights are considerably lower (less than 0.3 m) under the undisturbed current scenario (Figure 16A). At the landward edge of Laugharne Castle marsh, the hard defence prevents the propagation of waves, and thus Figure 16B shows a reduction in wave heights behind the defence. Other than that, wave height change in the estuary owing to hard defences is not significant. The impact of grazing (Figure 16C) causes a significant increase in wave heights, with the maximum value exceeding 0.1 m, across both Laugharne South and Laugharne Castle marshes, as well as on parts of Black Scar marsh. Compared with Figure 10C, which showed large-scale changes within the whole estuary, here, the differences are mainly on the marsh areas, and within the channel system north of Laugharne. For the easterly storm condition, the impact of managed realignment (Figure 16D) on wave heights within the estuary is less severe and very localised. There is an increase in wave heights as waves propagate into the Mylett farm realignment site, as well as small increases as waves propagate into the Mwche farm site.

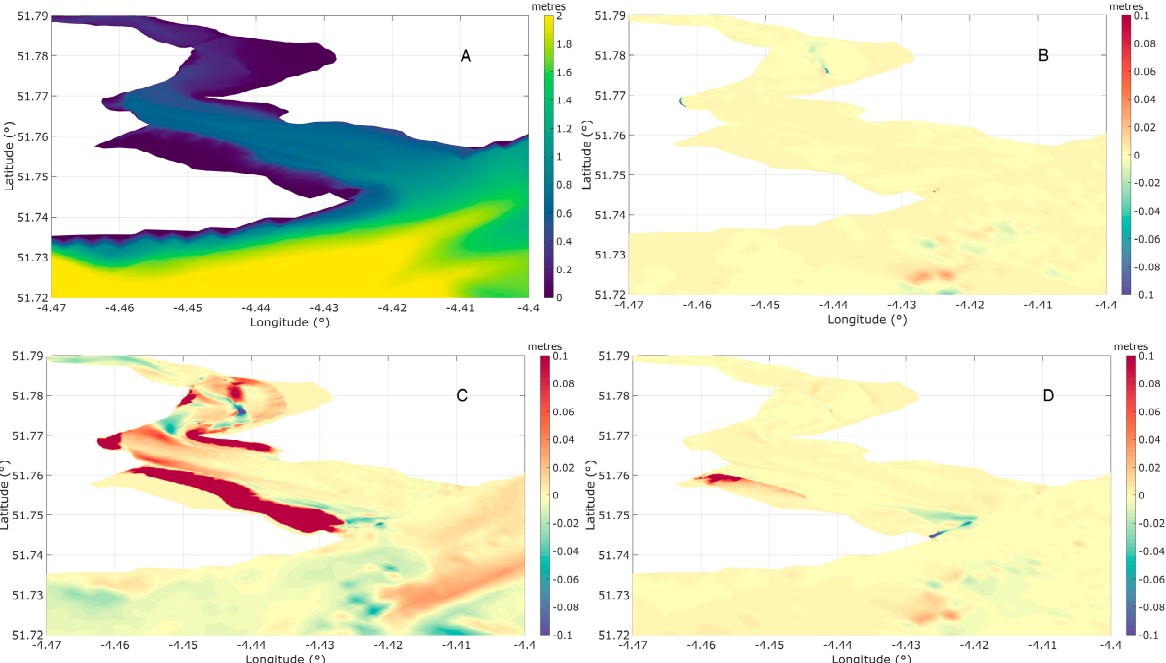

**Figure 16.** Comparison of peak storm wave height during the 1:50 year easterly storm. (**A**) Current condition, (**B**) difference between defended and current condition, (**C**) difference between grazed and current condition, and (**D**) difference between managed realignment and current condition.

The distribution of current velocity magnitude within the Taf Estuary, shown for the easterly storm condition in Figure 17, displays similar characteristics to that of the south-westerly storm condition (Figure 13). Currents in the Carmarthen Bay are higher than those in the estuary; however, owing to the lower total water level condition compared with the south-westerly storm condition, currents propagating along Pendine towards Ginst point are lower (~0.6 m/s). Flow within the two main channels and the connecting creek systems, which feed water over the saltmarshes, is proportionally lower (~0.3 m/s). Owing to the lower tidal extent within the estuary, there is limited flow over the marshes (0.1 m/s) compared with Figure 13. Again, hard defence has no widespread impact on currents.

For the grazed case, currents across all marsh areas are increased, although not to the same extent as Figure 13 owing to the reduced tidal extent. Again, this is synchronous with reductions in current magnitude in the channel system throughout the estuary (0.2 m/s). The managed realignment case does not cause such widescale differences. At the two breach locations, similar behaviour as displayed in Figure 13 is observed, although increases at the two breach sites are lower and to a lesser spatial extent. This is because of the lower water level condition, thus flooding less of the managed realignment sites.

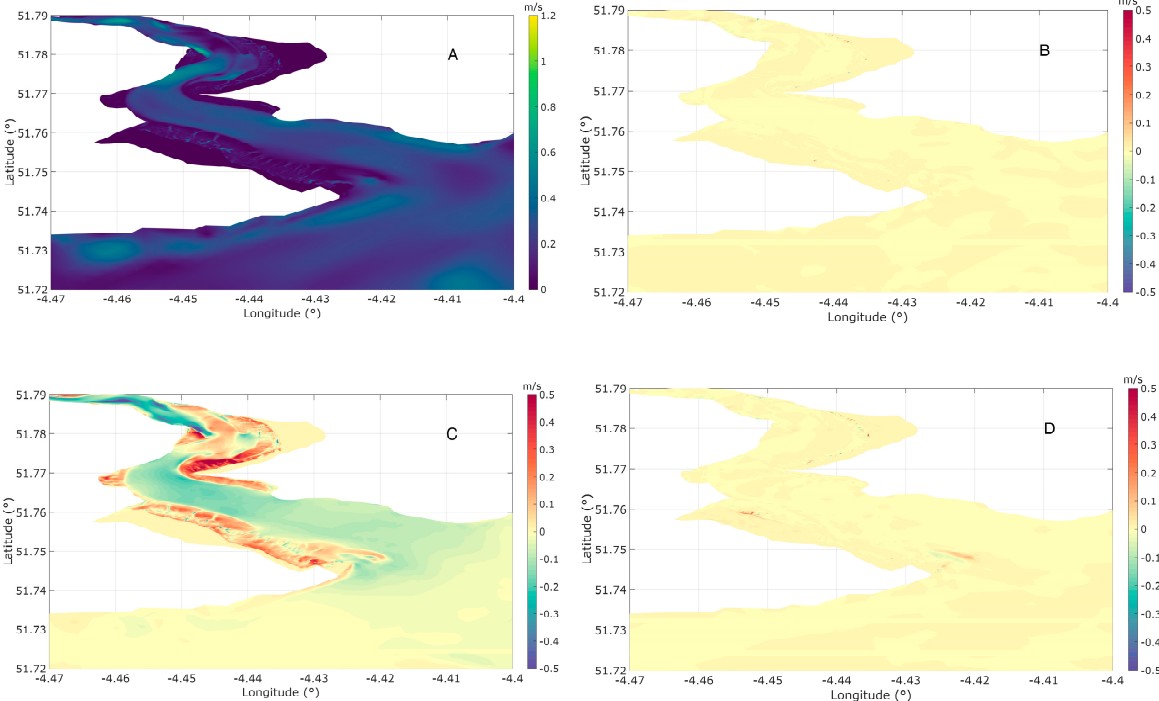

**Figure 17.** Comparison of peak flood current magnitude during the 1:50 year easterly storm. (**A**) Current condition, (**B**) difference between defended and current condition, (**C**) difference between grazed and current condition, and (**D**) difference between managed realignment and current condition.

Peak ebb currents during the easterly storm condition (Figure 18A) show a similar pattern to that of the south-westerly storm (Figure 14A). Although the strongest current velocities (~0.8 m/s) are limited to the two main tidal channels, high large velocities (~0.5 m/s) can be seen at numerous other places. As the total water level during the easterly storm is lower than that during the south-westerly storm, flow over the marshes is limited and, as such, the velocities on them are low (<0.1 m/s). As with Figures 13, 14 and 17, the impact of a defence showed no impact on flow velocities within the estuary. Compared with the south-westerly storm, the effect of grazing on flow velocities is lessened owing to the lower water levels. Increased flow velocities can be observed at the fringes of the marsh areas, but the largest differences occur towards the mouth of the estuary (0.5 m/s). The differences due to the managed realignment case are even more limited than in the south-westerly condition, because of the small volume of water entering the sites during flood tide.

As with the westerly storm, in order to investigate hydrodynamic change in the estuary and on the marshes due to management intervention, wave height change along the three cross sections T1, T2, and T3 (Figure 11) was examined in detail (Figure 19). In contrast with the south-westerly storm, the wave approach direction during easterly conditions is towards the Laugharne Castle and South marshes. In this case, wave heights at the seaward edge of all three marsh cross sections during the storm peak are higher than those of the westerly storm (0.6–0.7 m). Waves then rapidly dissipate on the marshes. Management interventions other than grazing do not induce any significant changes to wave

dissipation along the selected cross sections. However, when the marsh is grazed, wave dissipation is significantly smaller than that of the other cases, thus allowing 0.3 m–0.4 m waves reaching the landward edge of the marsh.

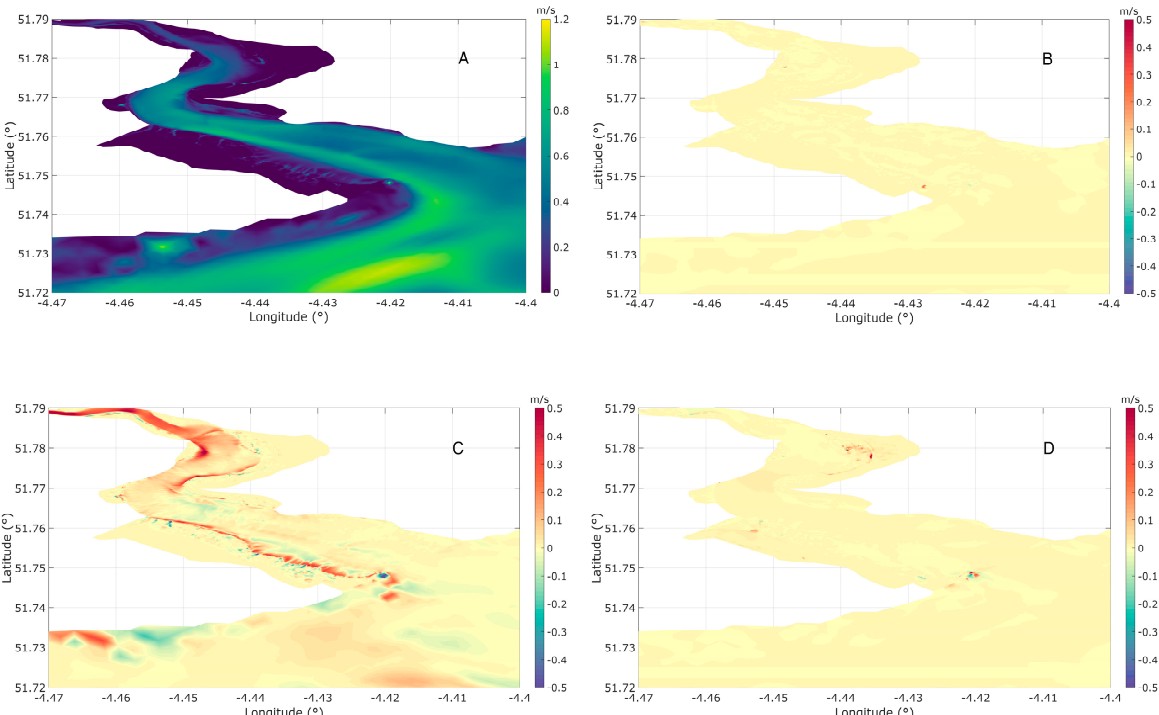

**Figure 18.** Comparison of peak ebb current magnitude during the 1:50 year easterly storm. (**A**) Current condition, (**B**) difference between defended and current condition, (**C**) difference between grazed and current condition, and (**D**) difference between managed realignment and current condition.

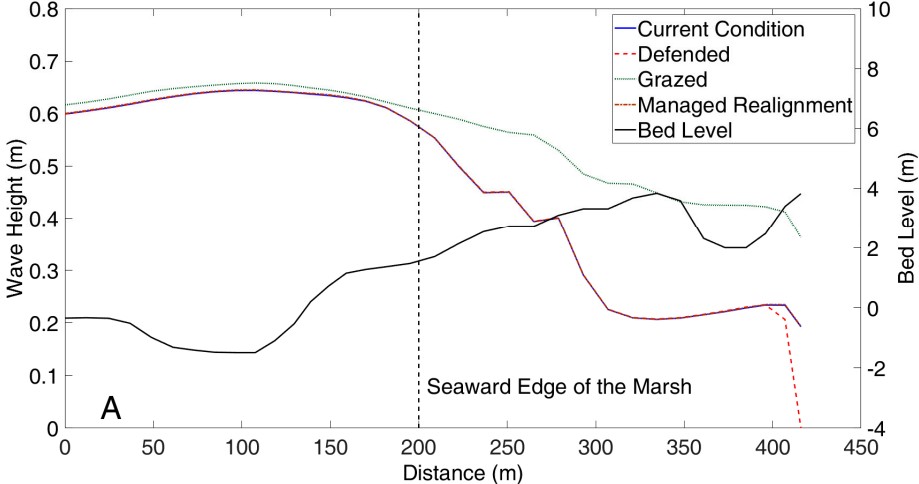

**Figure 19.** *Cont.*

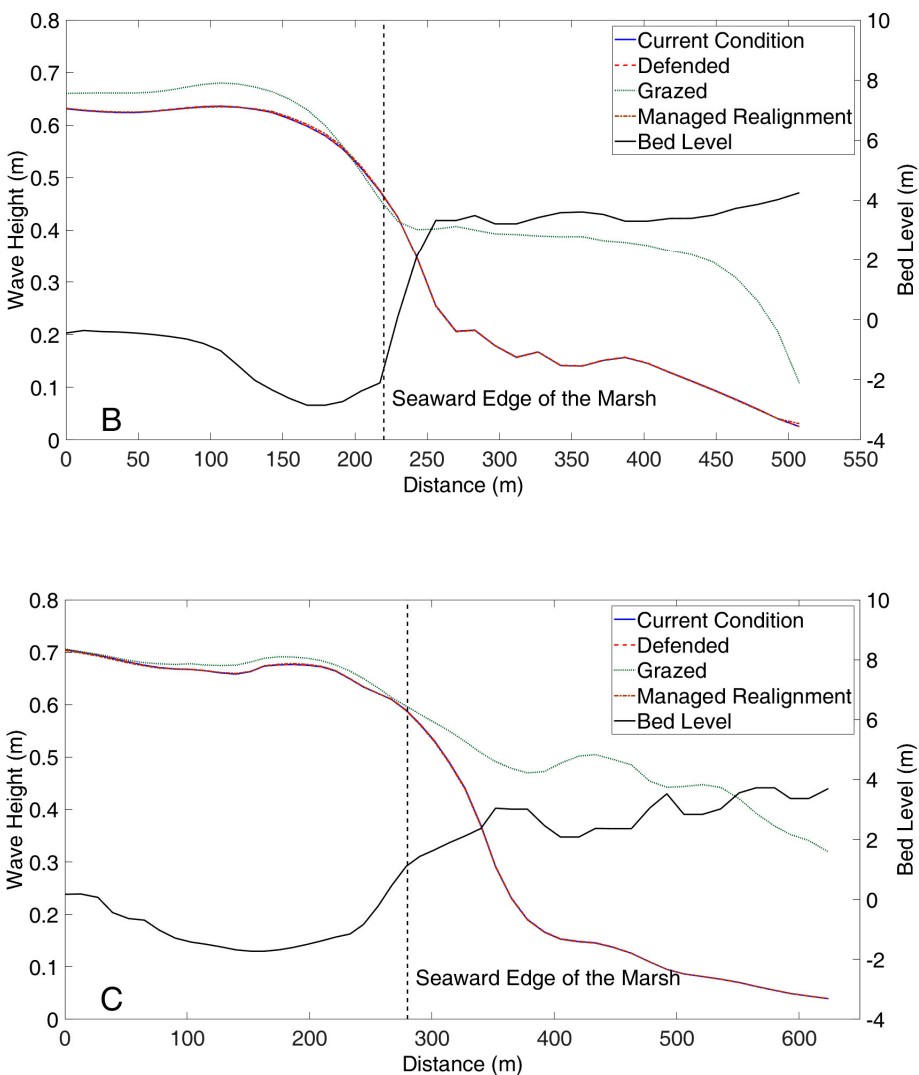

**Figure 19.** Comparison of peak storm wave height during the 1:50 year easterly storm at T1 (**A**), T2 (**B**), and T3 (**C**). Bed level in m OD. Please note, wave heights for 'current', 'defended', and 'managed-realigned' cases overlap with each other. Right hand side is the landward boundary.

## 5. Discussion

Some localised coastal management interventions, if implemented on saltmarshes of the Taf Estuary, will have notable impacts on the overall hydrodynamics of the estuary, in marsh areas in particular. It should be noted that the impact of the implementation of a hard flood defence at the Laugharne Castle marsh on the wider Taf Estuary may be of a very small scale. Although hard defences have been found to limit the storm surge attenuating effect by causing the setup of water levels against them [67], we do not see such an effect. This may be potentially owing to the size of the Laugharne Castle marsh and its position within the estuary. However, it should be noted that this study focuses only on the short-term impact of hard defences, although there may be some implications over a long period of time owing to long-term changes to sediment dynamics.

Although French [21] states that an increased tidal prism by managed realignment (e.g., seawall breach, thus allowing water to flow into previously dry areas) can modify tidal flow both locally and in the estuary as a whole, such a phenomenon was not identified here. While there is an obvious increase in water level at the two managed realignment sites, owing to the small increase in tidal prism relative to the current state of the estuary (0.9% [64]), the changes in tidal currents are limited to the areas at the close proximity to those two sites. This agrees with previous studies reported on the Blyth estuary in

the east coast of the United Kingdom that showed how restoration of tidal exchange to small areas leads to increased flows only in the vicinity of a seawall breach [31]. This may be linked to the relative increase in tidal prism and the position within the estuary, as French [31] found that the hydrodynamic effects of realignment were broadly in proportion to their incremental addition to the up-estuary tidal prism. Thus, although realignment sites do provide increased space for flood water owing to the storm surge, there are no notable impacts on water level in the rest of the estuary. However, it should be noted that Taf is a relatively small estuary with small-scale managed realignment. This observation may not hold true in larger estuaries with larger managed realignment schemes.

The differences in wave climate in response to coastal management interventions are more widespread for both south-westerly and easterly storms. Under south-westerly storms, locally generated waves within the estuary propagate from inside the Mylett farm realignment site, towards the breach and out into the lower estuary, thus increasing the wave heights from Laugharne to the mouth of the estuary. The impacts of easterly storms are less significant, potentially owing to the smaller water level due to small storm surge, limiting the potential for the generation and propagation of waves off the realignment sites. Therefore, the impact of the two managed realignment sites is limited.

Among all three interventions, grazing has the most widespread impact on the estuary for both storm conditions. The reduction of vegetation across the marsh platforms reduces the drag on the flow. As a result, flow velocities on the marsh platform are significantly higher than that of the current scenario. Counteracting the increase in flow velocities on the marshes, velocities in the channel were found to be smaller than that of the current scenario. The south-westerly storm has more influence on marshes located along the east bank of the estuary, while easterly storms have more influence on the marshes located along the west bank.

The impact of grazing on the wave climate shows a similar pattern. When comparing the grazed case with the current situation, it is evident that Laugharne Castle and South marshes limit the generation and propagation of waves within the estuary during the south-westerly storms. During easterly storms, they provide valuable wave attenuation function, thus reducing wave heights at the landward extent of the estuary. A summary of the effects of the various coastal management interventions on the hydrodynamics of the Taf Estuary is given in Table 5. Our study collectively suggests that excessive grazing of saltmarshes may leave some coastal areas surrounding the Taf Estuary more vulnerable to flooding. This raises concerns that grazing may alter the provisioning and diversity of ecosystem services provided by marshes [37], but it must be noted that we simulated an extreme over-grazing scenario and a more focused study is needed to understand the context dependency of this observation.

**Table 5.** A summary of the impacts of coastal management intervention options on waves and hydrodynamics of the Taf Estuary.

| Parameter | Intervention | | |
|---|---|---|---|
| | **Defended** | **Managed Realignment** | **Grazed** |
| Water Level | Reduction in water level behind defence, no widespread change | Increase in water level at both realignment sites, slight reduction in water level near breach locations | Small reduction in water level throughout the estuary |
| Wave Height | No change | Small-scale changes from waves generated in the managed realignment sites | Widespread differences with increases in wave height pronounced over marsh areas |
| Current | No change | Increases limited to the two breach sites and within the managed realignment sites | Currents increased over marsh areas, with a corresponding decrease in flow within the main channel |

As mentioned in Section 2, extreme surges in Carmarthen Bay coincide with extreme wind and wave events from the south-west. This generates large currents at the mouth of the Taf Estuary. There is limited wave propagation from the bay into the estuary owing to its orientation with respect to the

wave approach direction, and the shelter provided by the Ginst Point spit. As such, the waves in the estuary during south-westerly storms are mostly locally generated. On the other hand, the currents at the mouth of the estuary during easterly storms are lower, owing to the smaller peak storm water level during easterly storms. Wave heights at the mouth of the estuary are of a similar order of magnitude to those of the south-westerly storm, but they propagate into the estuary and onto the south western marshes. The location and configuration of marshes within the Taf Estuary, specifically Laugharne South and Laugharne Castle marshes, limit the generation and growth of waves within the estuary during easterly wind conditions. While the marshes on the opposite side of the estuary attenuate waves from the south west, when the wind is from the east, Laugharne South and Castle marshes attenuate the incoming waves.

As aforementioned, the interaction between changes in hydrodynamics and morphology may lead to developments in the estuary that may have further impacts upon coastal flooding and erosion beyond those highlighted. In addition, the threat of climate change on saltmarshes as well as the implication on coastal management practice discussed in Section 1 will lead to significant changes in estuarine hydrodynamics in the future. Regional sea level rise is predicted to be between 0.26 m and 0.67 m for representative concentration pathway (RCP) 2.6 and between 0.49 m and 1.1 m for RCP 8.5 in Carmarthen Bay in 2100, compared with the 1981–2000 average [68]. While here, the focus is on the short-term impact of management interventions, further study is needed in order to understand both the long-term development and management of the estuary under the impacts of climate change, and the potential role of management interventions under future climate conditions.

## 6. Conclusions

Saltmarshes are an integral part of most estuarine systems in the United Kingdom and elsewhere. They are also considered as natural buffers against flooding. In this paper, the effects of a selection of coastal management interventions on the hydrodynamic regime of a small macrotidal Taf Estuary are studied, providing insights into the role of saltmarshes in Welsh estuaries.

The effects of saltmarsh management interventions relate to several influencing factors. The increase in tidal prism following managed realignment at two saltmarsh sites, the position of the sites within the estuary, and the prevailing storm direction collectively contributed to some notable changes to the estuarine hydrodynamic regime beyond the saltmarsh sites. Although managed realignment is a preferred method for coastal management, the involvement of multiple factors and their potential impact on the wider estuary hydrodynamics prompt detailed studies of sediment and ecosystem dynamics, which are closely linked to estuarine hydrodynamics, before implementing such a scheme.

Although the establishment of a hard flood defence on the Laugharne Castle marsh prevented flooding in the village of Laugharne, it has no influence on the wider hydrodynamic regime of the estuary. However, hard defences, as a means of flood protection, are not popular among local communities owing to restricted access to the marsh and aesthetic issues.

Grazing of the marshes generated widespread changes in estuarine hydrodynamics. Although we considered the extreme case where marshes are grazed completely, which may not happen at any given time, the effect of grazing is important when considering widespread livestock grazing in most Welsh marshes. While the link between the changes in hydrodynamics and changes in morphology is complex and nonlinear, and was not the scope of this study, the increases in wave heights and current velocities across the marsh platforms highlight the increased marsh vulnerability as a result of excessive grazing. Therefore, grazing of saltmarshes will require careful management to minimise potential adverse impacts on flooding and marsh decline.

Saltmarshes on the east coast of England have been extensively studied, with several large managed realignment sites demonstrating the benefits of creating new saltmarsh habitats. Understanding the function of saltmarshes in Welsh estuaries, most of which are much smaller than the estuaries on the east coast, and are macro-tidal, is important to inform sustainable estuary management and

flood prevention. However, owing to the diverse nature of the Welsh coastline, and its large context dependency, inferring the characteristics of Welsh estuaries is complex.

Small, macro-tidal estuaries are relatively rare, and can be found around the United Kingdom and North Western Europe [69,70], as well as Japan [71], and provide unique environmental and socioeconomic value. This study provides useful information for making informed decisions on the selection and implementation of management intervention scenarios for small macro-tidal estuaries. However, it is important to extend the investigations to other similar estuaries to gain insights into context-dependency of these impacts.

Finally, the following limitations of the model and assumptions made in this study should be noted. (i) The study is limited to the worst-case scenarios, where extreme sea- and land-based hydrodynamic drivers coincide. (ii) Although the saltmarshes in the Taf Estuary consist of a variety of different species, the model assumed a uniform plant cover represented by a single drag coefficient. (iii) Potential impacts of marsh edge erosion during an extreme storm event were not considered as the model did not resolve morphology change.

**Author Contributions:** Conceptualization, W.G.B., T.J.v.V., T.P.F., J.N.G., and H.K.; methodology, W.G.B., T.J.v.V., T.P.F., J.N.G., and H.K.; software, W.G.B. and T.J.v.V.; validation, W.G.B., T.P.F., and T.J.v.V.; formal analysis, W.G.B., T.J.v.V., and H.K.; investigation, W.G.B., T.J.v.V., T.P.F., J.N.G., and H.K.; resources, T.P.F., H.K., and J.N.G.; data curation, W.G.B.; writing—original draft preparation, W.G.B., H.K., T.J.v.V., J.N.G., and T.P.F.; writing—review and editing, W.G.B., T.J.v.V., T.P.F., J.N.G., and H.K.; visualization, W.G.B. and T.J.v.V.; supervision, H.K. and J.N.G.; project administration, H.K. and J.N.G.; funding acquisition, H.K. and J.N.G. All authors have read and agreed to the published version of the manuscript.

**Funding:** This research formed part of the Valuing Nature Programme (valuing-nature.net), which is funded by the Natural Environment Research Council; the Economic and Social Research Council; the Biotechnology and Biological Sciences Research Council; the Arts and Humanities Research Council; and the Department for Environment, Food, and Rural Affairs. This research was supported by the UK Research Councils under Natural Environment Research Council award NE/N013573/1, Title CoastWEB: Valuing the contribution that COASTal habitats make to human health and WEllBeing, with a focus on the alleviation of natural hazards.

**Acknowledgments:** W.B. acknowledges the support of the Supercomputing Wales project, which is part-funded by the European Regional Development Fund (ERDF) via Welsh Government. CEFAS and the UK Met Office are acknowledged for providing wave and wind data. T.v.V. acknowledges the studentship supported by Swansea University to pursue his PhD studies. The paper contains public sector information, licensed under the Open Government Licence v3.0, from the Maritime and Coastguard Agency. The authors thank Suzie Jackson of Bangor University for providing ADCP data.

**Conflicts of Interest:** The authors declare no conflict of interest.

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
