# Peer review of "Computational Modelling of the Impacts of Saltmarsh Management Interventions on Hydrodynamics of a Small Macro-Tidal Estuary"

_jmse, doi:10.3390/jmse8050373_

Round 1
Reviewer 1 Report
The authors have investigated the impact of three fundamentally different flood mitigation intervention strategies on hydrodynamics of the Taf Estuary, located in the south-west Wales, UK. They applied a coupled FLOW-WAVE-Vegetation model, using the Delft3D coastal modelling software where the effect of saltmarsh vegetation on both the flow and waves was implemented. The computational modelling was performed for extreme storm events. The research is valuable as saltmarsh management interventions may potentially change hydrodynamic conditions in macro-tidal estuaries and increase the flood thread for low-lying areas adjacent to them. The article is interesting and worth publishing, however I have found many typing mistakes. Specific comments are given below.
Page 1, line 16, Abstract and many other examples: In the ms sometimes „Taf estuary” is written, sometimes „Taf Estuary”. In my view, if it is a geographical name, you should write it with a capital. The same I found for Carmarthen Bay (for example line 275 vs. line 252). Check all the geographical names in the ms and unify spelling.
Page 2, line 56 and many other examples: Rather than Doody, 2004 [16] write Doody [16]. Revise the ms for such mistakes.
Page 2, lines 75-92: The same as for line 56. Correct Friess et al., (2014); Townend and Pethick, (2011) showed; Pontee, (2015); Leggett et al. (2004). Erase the year of publication and insert the reference number in square brackets.
Page 2/3, lines 93-102: incorrect font style
Page 3, line 117: rather than “m3/s” write “3” using a superscript
Page 4, line 143 and many other examples: avoid two spaces between words.
Page 5, lines 184-198 (188, 196): rather than a year of publication use a reference number in square brackets.
Page 5, line 192 and page 6, line 213: Why was the Generalised Pareto Distribution used? Is it the best? Explain the cause of using it.
Page 6 lines 225 and 229: erase 2011.
Page 7 lines 238 and 253: erase 2011 and insert the reference number
Page 8, lines 266-267: You wrote that the Delft3D had been used successfully to investigate the ecology-hydrodynamic interface in saltmarsh systems. I wonder if these saltmarsh systems were similar or different from the Taf Estuary.
Page 8, line 276 vs line 279: again “Taf Estuary” or “Tal estuary”
Page 9, line 306: erase a year of publication
Page 10, table 3: In my view, the phrase in brackets should be under the table. Moreover erase years of publication.
Page 10, line 358: Does the citing literature [60] regard the value of NSE or the NSE itself? Precise in the text.
Page 12, lines 390 and 397; page 13 lines 409 and 416: erase years of publication and insert reference numbers
Page 19, line 556, page 20; line 601, page 21; line 623, page 22; line 631 and many other examples: “south-westerly storm conditions” , “easterly storm”, “easterly storm conditions” and other similar expressions sometimes but not always are written with a capital. Why? Check and unify, please
Page 23/24, lines 674-687: erase years of publication and insert the reference numbers
Page 23, line 680: “estuary” or “Estuary” with geographical names? And many other examples until conclusions (lines 708, 715, 730, 732, etc.)
Author Response
Page 1, line 16, Abstract and many other examples: In the ms sometimes „Taf estuary” is written, sometimes „Taf Estuary”. In my view, if it is a geographical name, you should write it with a capital. The same I found for Carmarthen Bay (for example line 275 vs. line 252). Check all the geographical names in the ms and unify spelling.
Response: m/s has been checked and revised to provide consistent capitalisation spelling for all geographical names
Page 2, line 56 and many other examples: Rather than Doody, 2004 [16] write Doody [16]. Revise the ms for such mistakes.
Response: m/s has been checked and revised to remove all such occurrences.
Page 2, lines 75-92: The same as for line 56. Correct Friess et al., (2014); Townend and Pethick, (2011) showed; Pontee, (2015); Leggett et al. (2004). Erase the year of publication and insert the reference number in square brackets.
Response: m/s has been checked and revised to remove all such occurrences.
Page 2/3, lines 93-102: incorrect font style
Response: Font style has been corrected.
Page 3, line 117: rather than “m3/s” write “3” using a superscript
Response: Use of superscript has been corrected.
Page 4, line 143 and many other examples: avoid two spaces between words.
Response: All erroneous occurrences of double spaces have been removed.
Page 5, lines 184-198 (188, 196): rather than a year of publication use a reference number in square brackets.
Response: m/s has been checked and revised to remove all such occurrences.
Page 5, line 192 and page 6, line 213: Why was the Generalised Pareto Distribution used? Is it the best? Explain the cause of using it.
Response: Added supporting statement and references for the use of the GPD “due to its successful application on a range of oceanographic variables [47,48]” lines 203-204.
Page 6 lines 225 and 229: erase 2011.
Response: 2011 has been removed
Page 7 lines 238 and 253: erase 2011 and insert the reference number
Response: 2011 has been removed and reference number inserted.
Page 8, lines 266-267: You wrote that the Delft3D had been used successfully to investigate the ecology-hydrodynamic interface in saltmarsh systems. I wonder if these saltmarsh systems were similar or different from the Taf Estuary.
Response: The estuarine systems modelled in most studies reported in literature are larger than our case study. However, some of them have similar marsh conditions, exposure and wave conditions.
Page 8, line 276 vs line 279: again “Taf Estuary” or “Tal estuary”
Response: m/s has been checked and revised to provide consistent capitalisation spelling for all geographical names
Page 9, line 306: erase a year of publication
Response: year of publication has been removed
Page 10, table 3: In my view, the phrase in brackets should be under the table. Moreover erase years of publication.
Response: Phrase in brackets has been moved to the footer and years of publication have been removed.
Page 10, line 358: Does the citing literature [60] regard the value of NSE or the NSE itself? Precise in the text.
Response: Citation has been moved to clarify the use of NSE itself.
Page 12, lines 390 and 397; page 13 lines 409 and 416: erase years of publication and insert reference numbers
Response: Years of publication have been removed and reference numbers inserted.
Page 19, line 556, page 20; line 601, page 21; line 623, page 22; line 631 and many other examples: “south-westerly storm conditions” , “easterly storm”, “easterly storm conditions” and other similar expressions sometimes but not always are written with a capital. Why? Check and unify, please
Response: Capitalizations have been removed from these expressions where unnecessary.
Page 23/24, lines 674-687: erase years of publication and insert the reference numbers
Response: Years of publication have been removed and reference numbers inserted.
Page 23, line 680: “estuary” or “Estuary” with geographical names? And many other examples until conclusions (lines 708, 715, 730, 732, etc.)
Response: All uses of estuary in geographical names have been capitalized, otherwise estuary is non capitalized
Reviewer 2 Report
Computational modelling of the impacts of saltmarsh management interventions on hydrodynamics of a small macro-tidal estuary
Bennett et al.,
General comments
The manuscript is focused on studying the hydrodynamic impacts of management interventions in comparison to the current ‘no intervention’ scenario and their impacts on wider estuarine hydrodynamics in a hypothetical extreme storm and river discharge events in the estuary. The results are interesting and useful though it can be further improved to have practical policy development with sustainable management implications. The authors should address the following aspects to further clarify and improve the manuscript before its publication.
The abstract should be improved so that it can be understood abstractly.
Under the introduction, the objectives of the study should be clearly stated while identifying the scenarios investigated using numerical modelling.
Under methodology, hydrodynamic parameters related to extreme event scenarios and sediment parameters (if sediment transportation is also modelled) should be given in a table.
Authors have focused only on the changes to the hydrodynamics of the estuary due to the management interventions and under two extreme wave conditions and the worst river discharge condition. Authors should rationalise the occurrence of two extreme events (coastal storm and maximum river discharge) simultaneously.
As a worst-case scenario, it is also better to superimpose extreme storm conditions with maximum spring high tide event.
Furthermore, extreme wave conditions change the morphodynamics and subsequently impacting hydrodynamics of the estuary.
Furthermore, authors have not considered the impacts of sea-level rise though its importance was emphasised in the Introduction. Though the results of the study are useful, management implications of the study are constrained to a certain extent due to the aforementioned limitations of the methodology.
Maybe such aspects can be discussed in a discussion. It may be interesting if the study will be extended to project the impacts of extreme events in the context to three management interventions and sea-level rise scenarios at least for 20 years.
Possible reasons for not changing in the flow velocities in the estuary or on the marsh areas during the south-westerly storm events in the Carmarthen bay or possible limitations of the approach have to be identified to obtain such results (Line 507-508). On the whole, model limitations and possible errors due to generalization and simplifications of the estuarine and saltmarsh processes have to be discussed.
Specific Comments
Line 25-27: Extensive grazing of saltmarsh was found to have the largest impact on wave heights, with increases in wave height over the marsh areas greater than 100% during easterly storm conditions.
Is the grazing by animals used as a planned management intervention as a mitigating or adaptation measure to any environmental issue or is it a human intervention with the ecosystem? This is confusing as the abstract explains the impacts of management interventions on hydrodynamics. The term” management interventions” implies that planned actions implemented to address the specific issue of the coastal system due to human or natural forcing. This clarification is particularly important as the abstract will stand in isolation of the main body of the text of the manuscript.
Line 30-33: The conclusion is not supported well with the result. An increase of current velocity and wave heights are pressures due to the anthropogenic interventions (drivers). The impacts, for instance, would be an increase in erosion and an increase in the area subjected to coastal flooding and overwash. It is important to identify the negative consequences of the increase in currents and waves. Such consequences will justify a better understanding of the changes (e.g. hydrodynamic and morphodynamic) in the coastal system due to management interventions.
Without considering the strict meaning of impact under the DPSIR analysis (Drivers, Pressures, State. Impact and Response), the word “impact” can be used in a general sense but rationalize the need for studies.
Line 51-52: Not clear. Loss of what? The sentence is not clear. Did authors mean that the loss of saltmarsh area and alteration of saltmarsh habitats around the world? Write it clearly.
Line 59-60: Give specific examples (national/regional/local) of Shoreline Management Plans that are already implemented or under consideration to implement in the study area.
Line 61-62: Authors talk about sea-level rise several times, but they have not taken impacts of sea-level rise into account when modelling the impact of management interventions on the hydrodynamics of the estuary.
Line 99-100. Here, grazing is considered not as management intervention but simply an anthropogenic intervention in the study area and that may need management interventions to address any of its environmental issues.
Line 103-105. Under objectives, name three different flood mitigation intervention strategies that were investigated in this study.
Line 110-119: Identify the location of the tidal currents, tidal heights, and river discharge. For instance, tidal height at the estuary mouth.
Line 143-145: Is the accretion and progradation of the coastline supported by coastal defence structures or is it exclusively due to the introduction of saltmarsh vegetation?
Line 162-177: Show flood defences in figure 2.
Line 178-181: Identify management intervention scenarios considered in this study.
Line 420-428: Is grazing implemented as a planned management intervention strategy in this study area?
Author Response
The abstract should be improved so that it can be understood abstractly.
Response: The abstract has been revised taking in to account several other comments, and condensed in to a more conceptual form. (lines 14-30)
Under the introduction, the objectives of the study should be clearly stated while identifying the scenarios investigated using numerical modelling.
Response: At the end of the introduction the three scenarios are detailed alongside the objectives of the study. Lines 108-111
Under methodology, hydrodynamic parameters related to extreme event scenarios and sediment parameters (if sediment transportation is also modelled) should be given in a table.
Response: No sediment transport is modelled, a new table providing a summary of the extreme event scenario parameters has been added to section 3.1.3
Authors have focused only on the changes to the hydrodynamics of the estuary due to the management interventions and under two extreme wave conditions and the worst river discharge condition. Authors should rationalise the occurrence of two extreme events (coastal storm and maximum river discharge) simultaneously.
Response: Although it is be possible that the maximum river discharge may not coincide with extreme sea condition, we focused on the potential worst case scenarios in this study. However, it should be noted that the river discharge in Taf is very small compared to the tidal prism of this macro-tidal estuary. Therefore, river discharge is not a critical parameter. New text is added to Section 3.2 to this effect.
As a worst-case scenario, it is also better to superimpose extreme storm conditions with maximum spring high tide event.
Response: As noted in the text, the guidance of MacMillan et al. (2011) has been followed in the combination of the surge profile with the base water level event to form the extreme water level profiles. Following the guidance this predicted tide should be between HAT and MHWS and represent a larger than 'normal' event but also to reach an appropriate level which reflects an event that occurs every year.
Furthermore, extreme wave conditions change the morphodynamics and subsequently impacting hydrodynamics of the estuary.
Response: The feedback loop between hydrodynamics and morphodynamics is acknowledged in the discussion section at two points (lines 699-701 and 755-758). As the focus here is on estuarine hydrodynamics, and due to the complexities and uncertainties associated with morphological modelling it is not discussed in further detail.
Furthermore, authors have not considered the impacts of sea-level rise though its importance was emphasised in the Introduction. Though the results of the study are useful, management implications of the study are constrained to a certain extent due to the aforementioned limitations of the methodology.
Maybe such aspects can be discussed in a discussion. It may be interesting if the study will be extended to project the impacts of extreme events in the context to three management interventions and sea-level rise scenarios at least for 20 years.
Response: Regional sea level rise for South Wales by 2040 is estimated to be approximately 0.1 m depending on the RCP selected compared with present conditions. This increase was not considered significant to investigate considering that extreme sea level values provided by McMillan et al. (2011) are only accurate to 1dp.
It is acknowledged that the study focuses on the short term impacts of the selected interventions (lines 699-701), discussion of the impact of sea level rise has been added to the discussion section, highlighting the need for further study of interventions in future (lines 755-764). To investigate long term responses, numerical modelling of the Taf Estuary which includes estuarine morphodynamic change, is required, which is outside the scope of this study.
Possible reasons for not changing in the flow velocities in the estuary or on the marsh areas during the south-westerly storm events in the Carmarthen bay or possible limitations of the approach have to be identified to obtain such results (Line 507-508). On the whole, model limitations and possible errors due to generalization and simplifications of the estuarine and saltmarsh processes have to be discussed.
Response: A discussion on potential reasons for changes in hydrodynamics, and in particular flow velocities can be found in Section 5, lines 704-711 and 723-728 of the m/s. A discussion on model limitations has been added to the ‘Conclusions’ (lines 802-807).
Specific Comments
Line 25-27: Extensive grazing of saltmarsh was found to have the largest impact on wave heights, with increases in wave height over the marsh areas greater than 100% during easterly storm conditions.
Is the grazing by animals used as a planned management intervention as a mitigating or adaptation measure to any environmental issue or is it a human intervention with the ecosystem? This is confusing as the abstract explains the impacts of management interventions on hydrodynamics. The term” management interventions” implies that planned actions implemented to address the specific issue of the coastal system due to human or natural forcing. This clarification is particularly important as the abstract will stand in isolation of the main body of the text of the manuscript.
Response: The abstract has been reworded reflecting that management of the use of land (lines 18-19), in particular allowing the use of saltmarsh area for grazing is a potential mitigation measure for flooding. Lines 24-25.
Line 30-33: The conclusion is not supported well with the result. An increase of current velocity and wave heights are pressures due to the anthropogenic interventions (drivers). The impacts, for instance, would be an increase in erosion and an increase in the area subjected to coastal flooding and overwash. It is important to identify the negative consequences of the increase in currents and waves. Such consequences will justify a better understanding of the changes (e.g. hydrodynamic and morphodynamic) in the coastal system due to management interventions.
Without considering the strict meaning of impact under the DPSIR analysis (Drivers, Pressures, State. Impact and Response), the word “impact” can be used in a general sense but rationalize the need for studies.
Response: We agree the literal use of the word ‘impact’ may lead to mis-understandings and mis-interpretations. Therefore, we replaced the word ‘impact’ by ‘effect’. As stated in Section 1, the focus of this study is to understand the effects of anthropogenic saltmarsh interventions for flood and coastal erosion prevention purposes on the hydrodynamics of the wider estuary system. Our conclusions are based on this context. The conclusions were reworded for clarity.
Line 51-52: Not clear. Loss of what? The sentence is not clear. Did authors mean that the loss of saltmarsh area and alteration of saltmarsh habitats around the world? Write it clearly.
Response: Sentence has been clarified to clearly state loss of saltmarsh area and alteration of saltmarsh habitats. Lines 48-49.
Line 59-60: Give specific examples (national/regional/local) of Shoreline Management Plans that are already implemented or under consideration to implement in the study area.
Response: The description of the Shoreline Management Plan guidelines for the study area within section 2 provides detail on the specific nature of the SMP policy within the Taf estuary. The references related to each management intervention scenario is given in Section 3.5 of the m/s.
Line 61-62: Authors talk about sea-level rise several times, but they have not taken impacts of sea-level rise into account when modelling the impact of management interventions on the hydrodynamics of the estuary.
Response: It is acknowledged that the study focuses on the short term (up to a decade) impacts of the selected interventions (lines 699-701), discussion of the impact of sea level rise has been added to the discussion section, highlighting the need for further study of interventions in future (lines 755-764). To investigate long term responses, numerical modelling of the Taf Estuary which includes estuarine morphodynamic change, is required, which is outside the scope of this study.
Line 99-100. Here, grazing is considered not as management intervention but simply an anthropogenic intervention in the study area and that may need management interventions to address any of its environmental issues.
Response: Widespread grazing is taking place in the saltmarshes of the Taf Estuary. It is widely known that grazing reduces plan height and density which in turn decreases wave attenuation as referenced in Section 3.5.2. However, there is no grazing management in Wales. Therefore, it is considered as a flood management option in our study, upon the request of Natural Resources of Wales. Text has been clarified to reinforce whether land is used for grazing or not is a management decision which can have implications for coastal protection. Lines 98-100.
Line 103-105. Under objectives, name three different flood mitigation intervention strategies that were investigated in this study.
Response: The three flood mitigation interventions are listed below the objectives Lines 103-105.
Line 110-119: Identify the location of the tidal currents, tidal heights, and river discharge. For instance, tidal height at the estuary mouth.
Response: Text has been clarified to identify the locations for each of these. Lines 112-122.
Line 143-145: Is the accretion and progradation of the coastline supported by coastal defence structures or is it exclusively due to the introduction of saltmarsh vegetation?
Response: The authors acknowledge the lack of studies and data to pinpoint the exact cause for the increase in saltmarsh area in recent decades. Anecdotally it seems as though the propagation of Ginst Point has created a more sheltered environment within the estuary that has promoted the growth of saltmarsh, which is qualitatively proved by historic photographs, but there is a lack of evidence to support this. It should be noted that our study is limited to the estuarine hydrodynamics but not the morphological change of the open coast of Camarthen Bay.
Line 162-177: Show flood defences in figure 2.
Response: Historic flood defences mentioned in this section have been added to figure 2.
Line 178-181: Identify management intervention scenarios considered in this study.
Response: Added a summary of the management intervention scenarios being considered. Lines 183-185
Line 420-428: Is grazing implemented as a planned management intervention strategy in this study area?
Response: Widespread grazing is taking place in the saltmarshes of the Taf Estuary. It is widely known that grazing reduces plan height and density which in turn decreases wave attenuation as referenced in Section 3.5.2. However, there is no grazing management in Wales. Therefore, it is considered as a flood management option in our study, upon the request of Natural Resources of Wales. A cross reference to Figure 2 has been added to the description of the grazed scenario to emphasise that the current saltmarsh areas are included in the grazed scenario and not those added as part of the managed realignment intervention.
Reviewer 3 Report
Line 30: There is inconsistency in how you denote the units. You sometime did not use a space after the number and used a space in other occasions. For example compare line 30 to line 330. Please be consistent and use only one style.
Line 71: Remove the extra space after “sea,”.
Lines 93-102: The font used in this paragraph is different than the rest of the manuscript.
Line 117: Typo: “m3/s”.
Methodology: Please expand this section more. Is this a 2D or 3D simulation? If 3D, how many vertical levels are used? What is the minimum water depth for the model?
Line 347-348: I disagree with this sentence. I would not say there is an “excellent” agreement when it is obvious that the model depth is clearly deeper than the measured. Please provide statistics to support your claim of “excellence”.
Lines 357:358: It would be helpful if you could show the formulas you used in an Appendix.
Author Response
Line 30: There is inconsistency in how you denote the units. You sometime did not use a space after the number and used a space in other occasions. For example compare line 30 to line 330. Please be consistent and use only one style.
Response: The formatting of units has been made consistent throughout the m/s
Line 71: Remove the extra space after “sea,”.
Response: All erroneous occurrences of double spaces have been removed.
Lines 93-102: The font used in this paragraph is different than the rest of the manuscript.
Response: Font style has been corrected.
Line 117: Typo: “m3/s”.
Response: Use of superscript has been corrected.
Methodology: Please expand this section more. Is this a 2D or 3D simulation? If 3D, how many vertical levels are used? What is the minimum water depth for the model?
Response: Added clarification to lines 288-290 “For this study the 2D depth averaged version of Delft3D FLOW is utilised, solving the unsteady shallow water equations with the hydrostatic pressure assumption.”
Line 347-348: I disagree with this sentence. I would not say there is an “excellent” agreement when it is obvious that the model depth is clearly deeper than the measured. Please provide statistics to support your claim of “excellence”.
Response: The authors acknowledge disagreement in model depth within the text, referring to the comparison as good, and noting the issues with equipment that may have caused it. The use of ‘excellent’ is only in reference to the comparison of phase between the two data sets. This was clarified in m/s.
Lines 357:358: It would be helpful if you could show the formulas you used in an Appendix.
Response: Delft3D is a widely known, open source coastal modelling suite used worldwide. Therefore, we do not intend to include model equations in the paper as it will duplicate information published in numerous previous literature. The reader is referred to [51] and the Delft3D Manual which comprehensively describes the model formulation and physical descriptions. Any additional formulae used in our study are already described in the text.
Round 2
Reviewer 2 Report
Authors have taken sufficient effort to respond and modify the manuscript according to comments and suggestions. Therefore, I will recommend that the manuscript can be accepted for publishing.
Reviewer 3 Report
The authors addressed my questions sufficiently though they did not address my comment about the statistical equations they used.